# PERCEPTOGRAM: VISUAL RECONSTRUCTION FROM EEG USING IMAGE GENERATIVE MODELS

## ABSTRACT

In this work, we reconstruct viewed images from EEG recordings with state-of-the-art quantitative reconstruction performance using a linear decoder that maps the EEG to image latents. We choose latent diffusion guided by CLIP embedding as the primary method of image reconstruction as it is currently the most effective at capturing visual semantics. We also explore reconstruction results from a latent space of PCA and ICA components, which capture luminance- and hue-related information from the EEG. The linear model provides interpretable EEG features relevant for differentiating general semantic categories of the images. We create spatiotemporal semantic maps that reflect the temporal evolution of class-relevant semantic information over time.[1]

## 1 INTRODUCTION

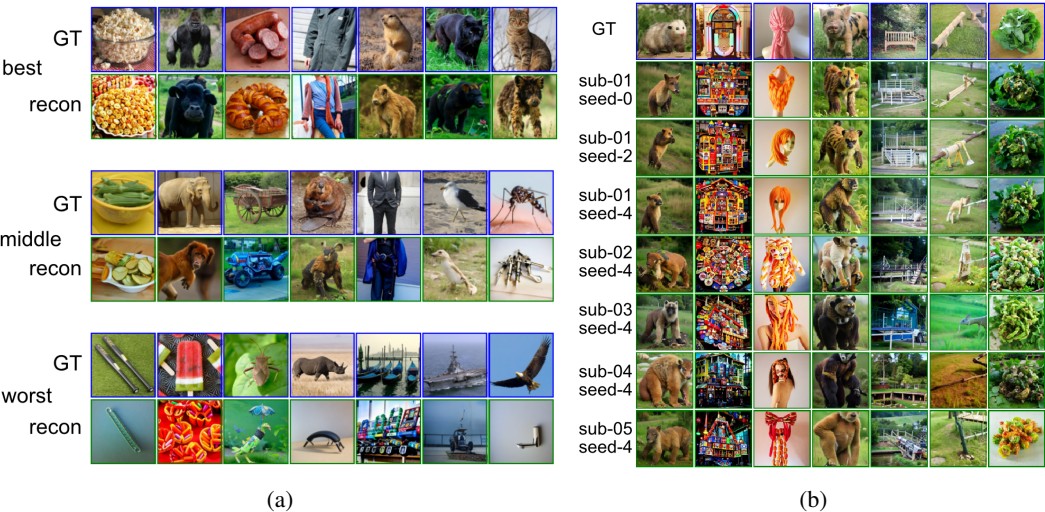

(a)                                                                                (b)

Figure 1: Reconstruction examples within and across subjects using our Versatile Diffusion Pipeline. (a) Example reconstructions of viewed images using EEG recorded from the viewing participant. Examples of the best, middle, and worst reconstructions were selected by visual inspection from Subject 1. The rows labeled GT (ground truth) show the image that was shown to the participant. The rows labeled reconstructed (recon) show the image that was created by our system shown in Fig. 2. Examples are sampled from among the best (top) middle (middle) and worst (bottom) reconstructions observed. (b) Robust reconstructions across subjects and diffuser random seeds. Top Row: the ground-truth stimulus images; subsequent 3 rows: different random seeds for the same subject; Last 4 rows: robust reconstructions across subjects.

Visual perception is an important aspect of human cognition and a gateway to understanding more complex cognitive processes such as visual mental imagery and dream visualization. Multiple

---

[1]Anonymous GitHub link for code: `https://anonymous.4open.science/r/EEG-Image-Reconstruction-BE27`

neuroimaging modalities including functional magnetic resonance imaging (fMRI), magnetoen-cephalography (MEG), and electroencephelography (EEG) have been used for studying the brain's response to visual scenes, each with its own trade-off of spatial and temporal resolution.

fMRI has millimeter-level spatial resolution, and the voxel activations extend into the brain with precise localization. At such meso-scale recording, fMRI is able to inform theories of how information is spatially organized in the brain. Basic scientific studies about visual (Kay et al., 2008) and language representations (Mitchell et al., 2008) in the brain with fMRI laid the groundwork for impressive visual reconstructions seen recently (Takagi & Nishimoto, 2023). There is even evidence suggesting that the brain representations generalize robustly from visual perception to visual imagery with fMRI (Kneeland et al., 2024). However, while spatially rich, fMRI measures Blood-Oxygen-Level-Dependent (BOLD) signals, which have a slow effective time resolution of a few seconds preventing visualization of dynamic responses.

EEG on the other hand, has spatial resolution limited by volume conduction which has resulted in its underuse in visual representation and reconstruction. Despite this, EEG responses to low-level visual features have been observed, including visual field (Halliday & Michael, 1970; Jeffreys & Axford, 1972), color (Paulus et al., 1988) and contrast (Schechter et al., 2005). In addition, specific event-related potentials (ERPs) such as the N170 are sensitive to semantic visual categories such as faces (Taylor, 2002). Importantly also, EEG has much better (millisecond) temporal resolution, allowing for tracking the temporal dynamics of brain representations.

Taken together, the high temporal resolution and promising prior work on visual feature mapping, along with its low cost and portability, make EEG an appealing neuroimaging modality for studying the dynamics of visual reconstruction.

## 2 METHODS

### 2.1 DATASET

We used the preprocessed version of THINGS-EEG2 dataset[2] from Gifford et al. (2022), which has 17 posterior EEG channels compared to the 63 total channels in the raw dataset. The EEG was initially sampled at 1000Hz and down-sampled to 100Hz during the preprocessing. The only major filtering method applied during the preprocessing is Multi-Variate Noise Normalization (MVNN), which computes the covariance matrices of the EEG data (calculated for each time-point), and then averages them across image conditions and data partitions. The inverse of the resulting averaged covariance matrix is used to whiten the EEG data (independently for each session) (Gifford et al., 2022).

In the experiment, each image is presented for 100ms followed by a blank screen for 100ms before the next image. The image presentation order is pseudo-randomized across the entire image set. All 10 subjects view the same 16740 images, of which the same 200 images are test images.

Trials are extracted from $-0.2$s to $0.8$s relative to the onset of the stimulus. Each training image is shown 4 times, and each test image is shown 80 times. We averaged all trials for the same image (within subject) to form the final dataset. At 100Hz sampling rate, $-0.2$s to 0.8 seconds corresponds to 100 samples. We discarded the first 20 samples which correspond to $-0.2$s to 0s, leaving 80 samples times 17 channels or 1360 dimensions per image per subject. The final dimensions of the training data for each subject are (16540 images, 1360 features) for the training set, and (200 images, 1360 features) for the test set. Responses to random subsequent images will affect each trial because images are presented every 200ms, but their effects should be reduced during the averaging step.

### 2.2 MODEL ARCHITECTURE

Our Versatile Diffusion Pipeline adopted the Ozcelik & VanRullen (2023) method, originally used for reconstruction of viewed images from fMRI signals. To summarize the method, the image reconstruction is a 2-stage process: the first stage maps the brain signal onto the latent space of a variational auto-encoder (VAE), specifically Very Deep Variational Auto-Encoder (VDVAE) (Child, 2020), which provides a rough visual representation that is then passed to the diffusion model and

---

[2]THINGS-EEG2 dataset: `https://osf.io/anp5v/`

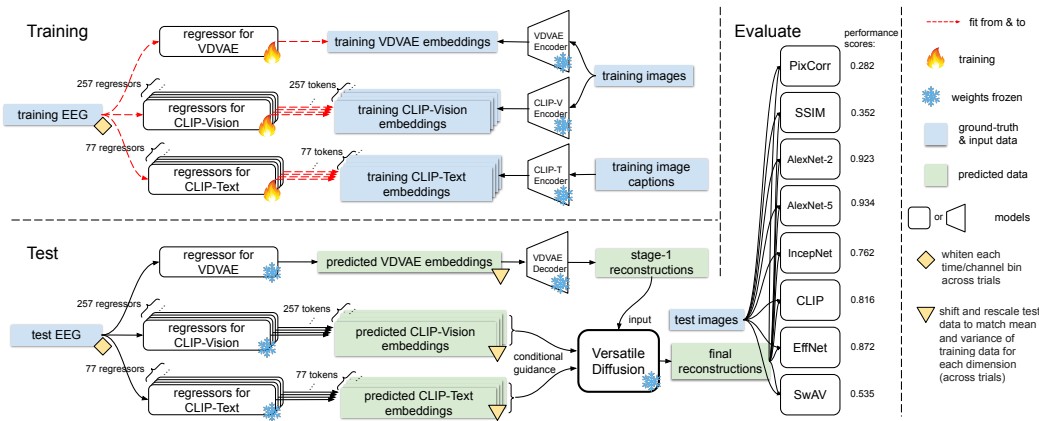

Figure 2: Flowchart illustrating the Versatile Diffusion variant of our reconstruction pipeline.

has a dimensionality of 91168 per image. The second stage maps the same brain signal onto each token of the CLIP-Vision (with dimensionality of 257 tokens by 768 dimensions per token) and CLIP-Text (with dimensionality of 77 tokens by 768 dimensions per token) embeddings of the Versatile Diffusion model (Xu et al., 2022), which combines the CLIP and the encoded images from the VDVAE and produces the reconstructed images.

The reconstruction pipeline consists of three stages: **training**, **testing**, and **evaluation** (refer to Fig. 2). In the **training** stage, images are processed through a VDVAE encoder, generating VDVAE embeddings (91168 dimensions), and through a CLIP-Vision encoder, producing CLIP-Vision embeddings (257 tokens × 768 dimensions). Corresponding captions are encoded by a CLIP-Text encoder to form CLIP-Text embeddings (77 tokens × 768 dimensions). Because CLIP-Vision and CLIP-Text embeddings have multiple tokens, separate regressors are trained to project the EEG data to each of the corresponding token space. Dashed red arrows indicate the fitting of these regressors from EEG to the embeddings, and only one arrow is shown for clarity since all regressors use the same EEG data. Before fitting, the EEG data is whitened to normalize each of the 1360 EEG dimensions across the 16540 training classes (refer to Appendix Section B for the mathematical definitions).

In the **test** stage, the predicted embeddings are shifted and rescaled to align their distributions with the training embeddings (refer to Appendix Section B for the mathematical definitions). Predicted VDVAE embeddings are fed into a VDVAE decoder to generate stage-1 reconstructions, capturing basic visual features such as shape and color. While the stage-1 VDVAE reconstructions primarily represent low-level visual features, the rich semantic details are preserved and guided by the predicted CLIP-Vision and CLIP-Text embeddings. These reconstructions serve as inputs to the Versatile Diffusion model, which incorporates guidance from the predicted CLIP-Vision and CLIP-Text embeddings to produce the final reconstructions.

In the **evaluation** stage, the final reconstructions are compared with the ground-truth images to assess performance.

For the THINGS-EEG2 dataset, the images come with their corresponding category names rather than a full-sentence description, so those category names were used to encode the training CLIP-Text embeddings. For VDVAE and CLIP-Vision, the original stimulus images are used to encode the training embeddings.

### 2.2.1 SPECIFIC IMPLEMENTATION DETAILS

For each image, each dimension of the 1360 dimensional signal is normalized across all training images. Then, one ridge regressor maps the 1360 dimensional signal onto the 91168 dimensional VDVAE embedding; 257 ridge regressors each map the same signal onto its corresponding token of the 768 dimensional CLIP-Vision embedding; 77 ridge regressors each map the same signal onto its corresponding token of the 768 dimensional CLIP-Text embedding. The regularization strength is

set to 1000 for the one VDVAE regressor, 1000 for all 257 CLIP-Vision regressors, and 10000 for all 77 CLIP-Text regressors.

We used 7 NVIDIA RTX A6000 48GB GDDR6 GPUs to run the reconstruction in parallel, though all reconstructions can be done on a single GPU as long as it has more than 13GB of graphics memory. Each 200 images of reconstruction takes around 15 minutes to complete. The ridge regression training is run on a CPU, and takes only a few minutes to complete. This highlights the compute efficiency of the linear models: excluding the diffusion process itself, the mapping from EEG to the predicted latents can be done on most home computers while maintaining good performance.

## 2.3 Performance Evaluation Metrics

We used the same performance metrics (see Fig. 3) as in Ozcelik & VanRullen (2023), which has been used in other followup studies such as MindEye Scotti et al. (2023). The 8 metrics we used are Pixel Correlation (PixCorr), Structural Similarity (SSIM), AlexNet layer 2 and 5 outputs pairwise correlations, InceptionNet output pairwise correlation, CLIP ViT output pairwise correlation, EfficientNet output distance, and SwAV output distance.

PixCorr and SSIM involve comparing the reconstructed image with the ground-truth (GT) test image. PixCorr is a low-level (pixel) measure that involves vectorizing the reconstructed and GT images and computing the correlation coefficient between the resulting vectors. SSIM is a measure developed by Wang et al. 2004 that computes a match value between GT and reconstructed images as a function of overall luminance match, overall contrast match, and a "structural" match which is defined by normalizing each image by its mean and standard deviation. We should note that this measure was designed for comparing images with minor distortions and does not seem as reliable for images that are not close matches as currently obtained with image reconstruction methods. One way to see this is to observe (in Fig. 3b) that the SSIM measure does not seem much affected by the duration of EEG window over 50ms, unlike the other measures that show performance improvements from 100ms through 400ms.

Fig. 3a is computed by reconstructing with Versatile Diffusion using 7 different random seeds. For each subject, the final performance is the average across the 7 runs. The standard deviation across the 7 runs for each subject is represented by the error bars.

Fig. 3b is computed by models trained on each subject's 4-trial-averaged training data, and tested on their corresponding 80-trial-averaged test data. For Fig. 3b, the "first 200ms", "first 400ms", "first 600ms" and "first 800ms" models use those corresponding time ranges after the onset of the stimulus. The 0ms performance, which should correspond to chance level, is computed by passing the 200ms before the onset of the stimulus onto the trained "first 200ms" model. The bars heights and the error bars represent the mean and standard deviation across the 4 subjects.

Fig. 3c is computed by gradually increasing the the number of training images and the number of averages in the test samples. In Fig. 3c, the y-axis shows gradual increase of the number of training images, and the x-axis shows gradual increase of the number of trial averages for each of the test images. Performance varies smoothly as a function of both training images and test trials.

The Versatile Diffusion Pipeline maps each trial onto 3 latent embeddings used for the reconstruction: VDVAE, CLIP-Vision, and CLIP-Text. To assess the relative contribution of the 3 kinds of latent embeddings toward the model performance we tested 6 kinds of ablated models (Fig. 3d): "VDVAE only" used the low-resolution reconstruction outputs of the VDVAE; "CLIP-Text only" uses only the CLIP-Text embedding; "no CLIP-Vision" uses VDVAE and "CLIP-Text"; "CLIP-Vision only" uses only the CLIP-Vision embedding; "no CLIP-Text" uses VDVAE and CLIP-Vision; "no AutoKL (VDVAE)" uses CLIP-Vision and CLIP-Text.

The ablation of the VDVAE is done by feeding a blank gray image (127, 127, 127 in RGB) instead of the results from VDVAE as the input image into Versatile Diffusion, and increasing the diffusion strength from 0.75 to 0.99. The ablation of the CLIP-Vision is done by encoding a blank black image as CLIP-Vision instead of using the predicted CLIP-Vision from EEG data and setting the mixing ratio from 0.4 to 0.99. (a blank black image is defined as the unconditioned image for this part). The ablation of the CLIP-Text is done by encoding an empty string as CLIP-Text instead of using the predicted CLIP-Text from EEG data and setting the mixing ratio from 0.4 to 0.

Table 1: Quantitative assessments of the reconstruction quality for EEG, MEG, and fMRI. For our algorithm we give the mean and standard deviation across 10 subjects with random seed 0. For detailed explanations of the metrics see section 2.3.

| Dataset ↑ | Low-level | | | | High-level | | | |
|---|---|---|---|---|---|---|---|---|
| | PixCorr ↑ | SSIM ↑ | AlexNet(2) ↑ | AlexNet(5) ↑ | Inception ↑ | CLIP ↑ | EffNet ↓ | SwAV ↓ |
| NSD (Brain-Diffuser) Ozcelik & VanRullen (2023) | 0.254 | 0.356 | 0.942 | 0.962 | 0.872 | 0.915 | 0.775 | 0.423 |
| NSD (MindEye) Scotti et al. (2023) | 0.309 | 0.323 | 0.947 | 0.978 | 0.938 | 0.941 | 0.645 | 0.367 |
| THINGS-MEG (B.D.) Benchetrit et al. (2024) | 0.088 | 0.333 | 0.747 | 0.855 | 0.712 | 0.804 | - | 0.576 |
| THINGS-MEG (EEGImageDecode) Li et al. (2024) | - | 0.340 | 0.613 | 0.672 | 0.619 | 0.603 | - | 0.651 |
| THINGS-EEG2 (EEGImageDecode) Li et al. (2024) | - | 0.345 | 0.776 | 0.866 | 0.734 | 0.786 | - | 0.582 |
| THINGS-EEG2 (ours) | $0.267 \pm .015$ | $\mathbf{0.347} \pm 0.003$ | $\mathbf{0.910} \pm 0.010$ | $\mathbf{0.927} \pm 0.005$ | $\mathbf{0.752} \pm 0.008$ | $\mathbf{0.807} \pm 0.009$ | $\mathbf{0.877} \pm 0.004$ | $\mathbf{0.540} \pm 0.004$ |

# 3 RESULTS

## 3.1 PERFORMANCE

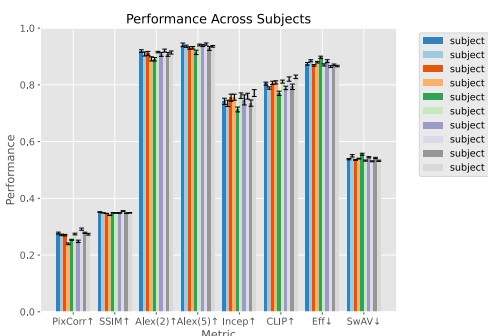

(a) Performance across 10 subjects.

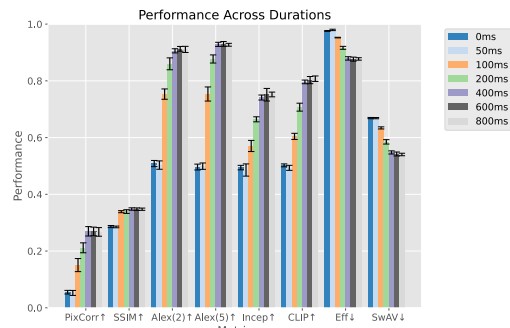

(b) Performance across durations for Subjects 1 through 4.

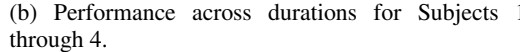

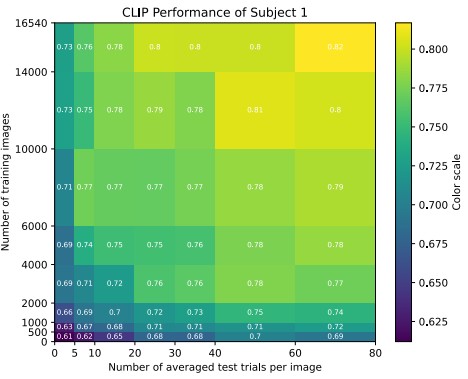

(c) CLIP performance across training sizes and number of test trial averages shown for Subject 1 with random seed 0 used for reconstructions.

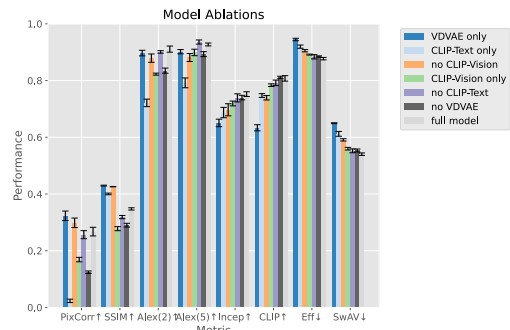

(d) Model ablation. Bar heights and error bars represent the mean and standard deviation across Subjects 1 through 4.

Figure 3: Basic Performance Metrics.

The reconstruction performance of our simple linear model (shown quantitatively in Table 1 and qualitatively in Figure 1a) beats the state-of-the-art's reconstruction performance on all metrics that they reported, which used a much more sophisticated, transformer-based decoding algorithm (Li et al., 2024). Our model also performed better on the EEG data than the neural network-based model by Benchetrit et al. (2024) on MEG data.

The performance across the 10 subjects is relatively consistent with a small amount of variation as shown quantitatively in Fig. 3a and qualitatively in Fig. 1b. Performance improves with increases in training data and repetitions of test data as shown in Fig. 3c. The performance across duration

shows that using 400ms of data achieves a slightly higher performance than 200ms, despite the fact that other images have started showing by this time (see Fig. 3b).

## 3.2 MODEL ABLATION

Fig. 3d shows that the full model achieved the best all-around performance compared to the ablated models. In general models without CLIP-Text achieved a similar level of performance and are slightly better than models without CLIP-Vision. The stage-1, VDVAE reconstructed images have good low-level performance, but poor high-level performance.

## 3.3 VDVAE, PCA, AND ICA RECONSTRUCTIONS

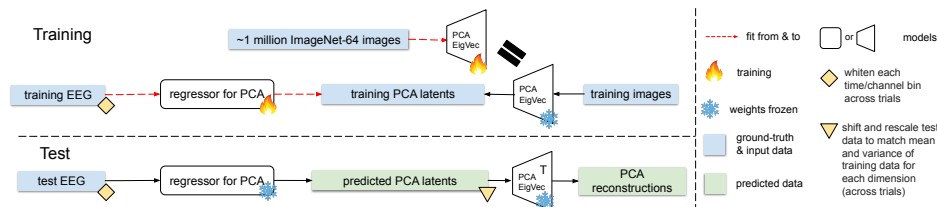

Figure 4: Flowchart illustrating the PCA reconstruction pipeline.

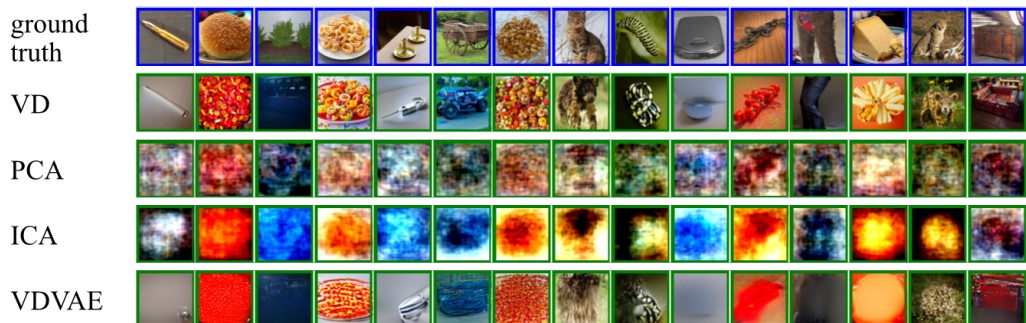

Figure 5: Reconstructions using different latent spaces. **GT**: ground truth stimulus images; **VD**: Versatile Diffusion reconstructions; **PCA**: PCA-based reconstruction; **ICA**: ICA-based reconstructions; **VDVAE**: VDVAE-based reconstructions.

In the Versatile Diffusion reconstruction pipeline, the VDVAE is used as stage-1 reconstruction and input to the Versatile Diffusion to better align the low-level visual features such as color, shape, and spatial frequency. We can visualize the VDVAE reconstructions separately to see these reconstructed features (see the row "VDVAE" in Fig. 5). Similarly, we can use the principal components or independent components of images as the latent embedding target space for the EEG mapping.

We fit a PCA model on ImageNet-64 dataset with around 1 million images (see flowchart in Fig. 4). We used the top 1000 resulting principal components to encode the THINGS-EEG2 training and test images to form the PCA latents. Then we trained a linear regressor to map from EEG to PCA latents. Finally, we reconstruct the images by using the standard PCA reconstruction algorithm.

The PCA reconstructions contains information about color, luminance, and some shape information (see row "PCA" of Fig. 5).

The ICA reconstruction pipeline (Fig. A.5) works similarly to the PCA pipeline and is also trained on ImageNet-64. The ICA reconstructions contains information about color in terms of cold versus warm, and some shape information (see row "ICA" of Fig. 5).

# 4 ADDITIONAL ANALYSES

## 4.1 SCALP-ASYMMETRY EFFECT

ground
truth

VD

mirrored

half

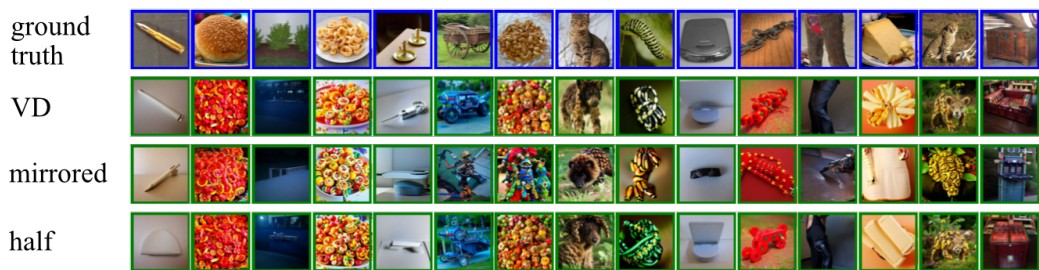

Figure 6: Reconstructions for different experimental manipulations. **ground truth**: ground truth stimulus images; **VD**: Versatile Diffusion reconstructions with all 17 posterior channels; **mirrored**: Versatile Diffusion reconstruction trained with all 17 posterior channels but all the electrode locations mirrored about the midline during test time (e.g. data from electrodes on the right scalp mapped to channels trained with data from electrodes on the left side); **half**: Versatile Diffusion reconstruction with half of the electrodes (8 electrodes: P7, P3, Pz, P4, P8, O1, Oz, O2) in the international 10-20 configuration (see Fig. 8a for locations).

As the right visual field maps to the left primary visual cortex, and the left visual field maps to the right primary visual cortex, an interesting experiment is to examine the effect of swapping electrode data after training. That is, after training we wish to explore the effect of putting the data from right side electrodes into channels trained on corresponding electrodes from the left scalp.

At the same time, mirroring the electrode location on the test EEG data caused significant degradation of the reconstruction performance and interestingly, often the "animal nature" of animal images disappeared (see for example the second last column in Fig. 6), though not always. Decreasing the channel density for the training and test EEG data caused much less drop in reconstruction performance (compare last two rows of Fig. 6). This indicates that the EEG representations of high-order visual features are somewhat spatially asymmetric across the midline.

## 4.2 TIME-SWAP EFFECT

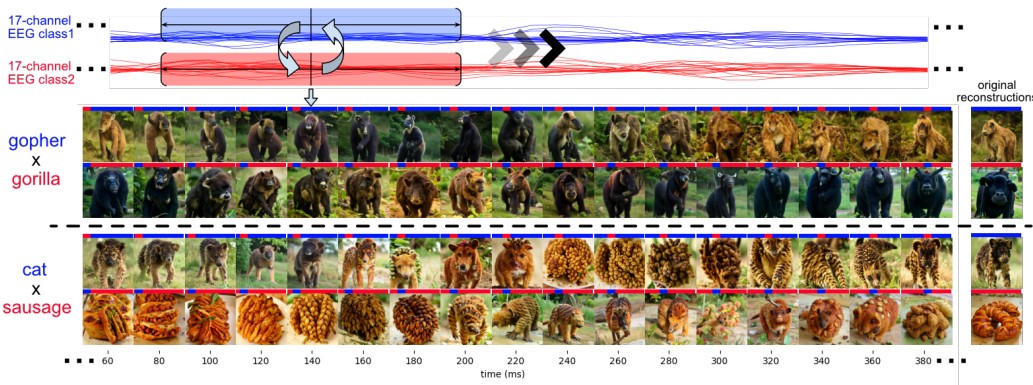

Figure 7: Illustration of our Time-Swapping Experiment. **Top**: illustrates time segment swapping as a sliding window with the down arrow pointing to the corresponding reconstruction; **Bottom**: bar color over each image illustrates proportionally which segments come from its own EEG and which comes from EEG to the other class.

In order to investigate the temporal dynamics and the salient features in the EEG data, we develop a novel technique to find time-ranges that are most sensitive to disturbance. We used pairs of images

and swapped analogous time segments of data between EEG responses to each of the images as demonstrated in Fig. 7.

Each image results from reconstruction of EEG where a 120ms time window centered at the corresponding time point is swapped between the 2 classes within that window while holding the signal outside the window the same. On top of each reconstructed image, we added a color bar that proportionally indicates which EEG time segment is swapped with the other class for that image. The two classes are represented by red and blue in this color bar and time is represented in the horizontal direction so a blue bar with a small red square represents that 120ms of the EEG at its relative location is swapped with the EEG for the other class. Notice how the small squares progress to the right as the samples progress to the right. The original, unswapped reconstructions (shown at right) have their color bars all blue/red, indicating that no part of their EEG is swapped with the other class.

In the gopher-gorilla swap experiment, the reconstructed "gopher" image has darker fur when the swapped windows are centered at 100ms through about 260ms (when 120ms time windows from 100-60=40ms to 260+60=320ms are replaced with the EEG to the gorilla from the corresponding time frame). Similarly the gorilla has a lighter fur color when the EEG in about the same time range is replaced with the EEG from the gopher presentation. In the cat-sausage swap experiment, the cat reconstruction has a food-like appearance when 120ms windows centered from 240-280ms and the sausage has an animal-like appearance when 120ms windows centered from 200-360ms are replaced with EEG from the cat presentation. The later sensitive time period for the semantic differences (animal vs. food) compared to the fur color differences (light vs. dark) reveals later processing of semantic compared to low-level visual features.

### 4.3 SPATIOTEMPORAL SEMANTIC MAP

While the reconstructions reflect semantic alignment, we would like to delve further into the relevant EEG features underlying the semantic classes. We did show that by mirroring the electrodes at test time the semantic alignment degrades significantly indicating that, for individuals, the spatial features underlying semantic category tend to be spatially asymmetric across the mid-line. In order to see how different electrodes signal different general semantic categories over time, we created a visual semantic map of viewing images adapted to the temporal resolution of EEG. Note this is inspired by the fMRI semantic map by Huth et al. (2016), but, critically, it is not a static map as in the fMRI case; it is a map that unfolds in time – a spatiotemporal map (see Fig. 9).

To create the map, we use a more efficient CLIP embedding and diffusion model unCLIP (Ramesh et al., 2022), which has only 1024 dimensions in the embedding space. (Reconstructions with unCLIP are similar to those with our original Versatile Diffusion Pipeline, see Fig. A.8). We trained a linear encoder model to re-encode the predicted CLIP-Vision embeddings back into EEG. The re-encoded EEG – here called "EEG patterns" (in analogy to common spatial patterns Blankertz et al. (2008) used in brain-computer interfaces) – accentuates features that are relevant for the visual semantics that CLIP-Vision embeddings capture (see Fig. A.9a for a subset of test images), compared to the real EEG (see Fig. A.12a for the full pattern and compared to real EEG Fig. A.12b). For better visualization, the 200 classes on the y-axis are already organized neatly into 3 general semantic groups (others, animals, food) derived by hierarchical clustering on the CLIP-Vision embedding vectors of the images. Within the "others" group (it can be further subdivided into "tools", "vehicles" and "clothing", but here for simplicity we simply group them as "others").

We averaged the rows in "food" and "animals" and "others" to form the averaged EEG pattern for these 3 general semantic groups, and plotted them on a 2-D topological scalp map (topomap) where dots on the map represents the actual locations of the electrodes on the scalp (see Fig. 8a). Figure 9 shows the result across time for the average of these maps over all subjects for the "food", "animals", and "others" groups. The spatiotemporal semantic map for Subject 1 is shown in Figure A.10 and representations for maps of all individuals in Figure A.11. There are clear patterns across people, but some individual differences in asymmetry and lateralization strength. The individual assymetries likely underlie the performance degradation we observed earlier from mirroring the electrodes.

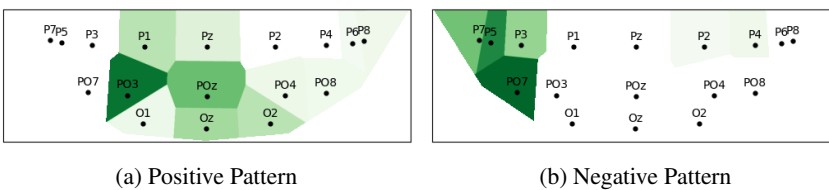

(a) Positive Pattern          (b) Negative Pattern

Figure 8: A temporal slice of Subject 1's spatiotemporal semantic map at 200ms for the animals category. The figure shows electrode locations at the back of the head using the standard 10/10 naming system. The positive pattern shows increased voltage relative to the grand average response. The negative pattern shows decreased voltage relative to the grand average response. Darker colors represent stronger differences.

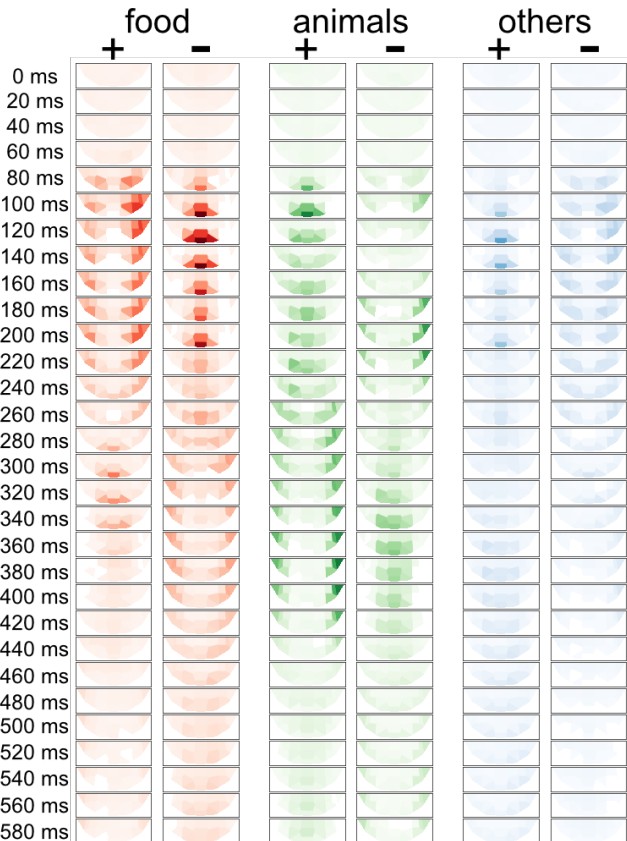

Figure 9: The spatiotemporal semantic map averaged across 10 subjects. We take the average across images in the same general category, and plot the patterns of the channels in topological maps across time. Each miniplot represents the cross-subjects-average activity over the back (visual) part of the brain and different time segments (rows). Each row from top to bottom represents a 20ms time window going from 0 to 600ms. We color the animal average green, food red, and everything else blue. A "+" indicates that category if there is a more positive voltage there; a "-" indicates that category if there is a more negative voltage there. (i.e. they reflect where increased positive and negative activity indicates the category). With this plot, we can see the most general dynamical patterns for 3 general categories and can easily spot some consistent general trends. (A result for an individual subject is shown in Fig. A.12a)

## 5 DISCUSSION

We have demonstrated the surprising power of learned linear mappings to different latent spaces. When mapped to the CLIP and VDVAE latent spaces and reconstructed with Versatile Diffusion, realistic reconstructions are obtained that numerically outperform previous EEG reconstruction attempts.

### TECHNICAL NOVELTY

We also demonstrated that even the principal components of images can be used as the latent embeddings for EEG mapping to reconstruct images; these are admittedly very blurry, but do contain surprising color and shape information. While PCA and EEG have existed for decades, no attempt to reconstruct EEG with image PCA has been made. This underscores the novelty of these findings for EEG.

### CONTRIBUTION TO COGNITIVE SCIENCE

Finally we created a spatiotemporal map of the dynamic EEG representation for animal, food, and other objects. There is a large amount of consistency between people, but some individual differences especially in relative lateralization.

### 5.1 LIMITATIONS

The dataset used in this work used fully randomized RSVP-style (rapid) presentation with stimuli lasting 100ms and new stimuli appearing every 200ms. In all the reconstruction work in the literature multiple repetitions of test data are required for high performance. (Note this is not dissimilar from standard cognitive neuroscience studies which also average responses to several repetitions).

Even though the linear model itself is fast and computationally efficient, the diffusion models for generating the images such as Versatile Diffusion and unCLIP still require dedicated desktop-level GPUs to compute. Using distillation based image generation models while maintaining the CLIP guidance aspect could reduce the amount of computation.

### 5.2 FUTURE DIRECTIONS

The EEG patterns for individual subjects here can serve as canonical patterns. Future research could try modifying the experimental paradigm to see if or how the newly acquired patterns from the modified experiment differ from the canonical patterns.

Future research may also work towards video reconstruction from EEG as AI generated video is approaching a similar level of sophistication as AI generated images. Video stimuli may be more representative of natural visual experiences. Motion-related features in videos may also contain decodable dynamics that would take advantage of the temporal resolution of EEG and MEG. Decoding a sub-section of a video clip could provide insight into mechanisms of ongoing visual processing, which, compared to evoked transient visual responses, might be more similar to internal visual representations such as visual imagery and visual dreams.

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

# A APPENDIX

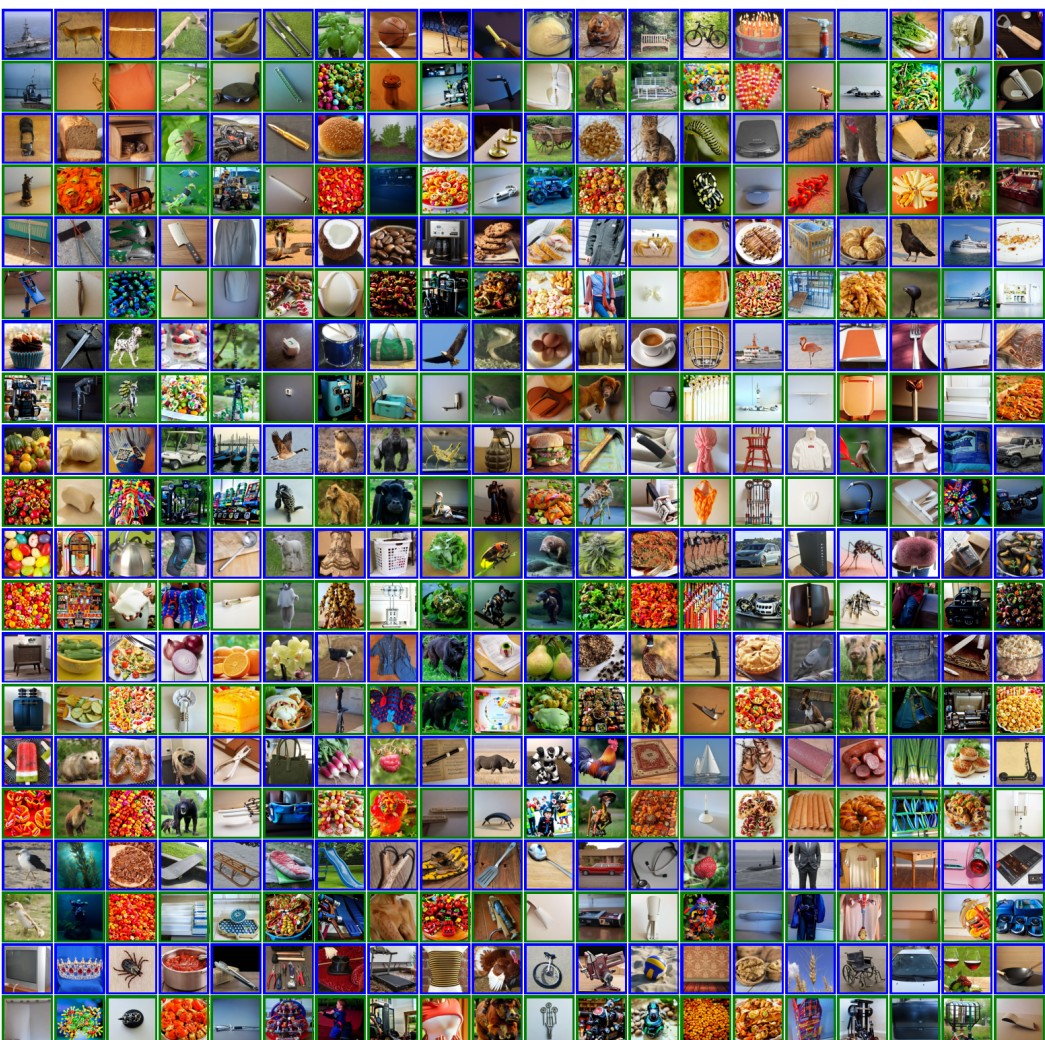

Figure A.1: Full reconstructions (Subject 1) using the Versatile Diffusion reconstruction pipeline.

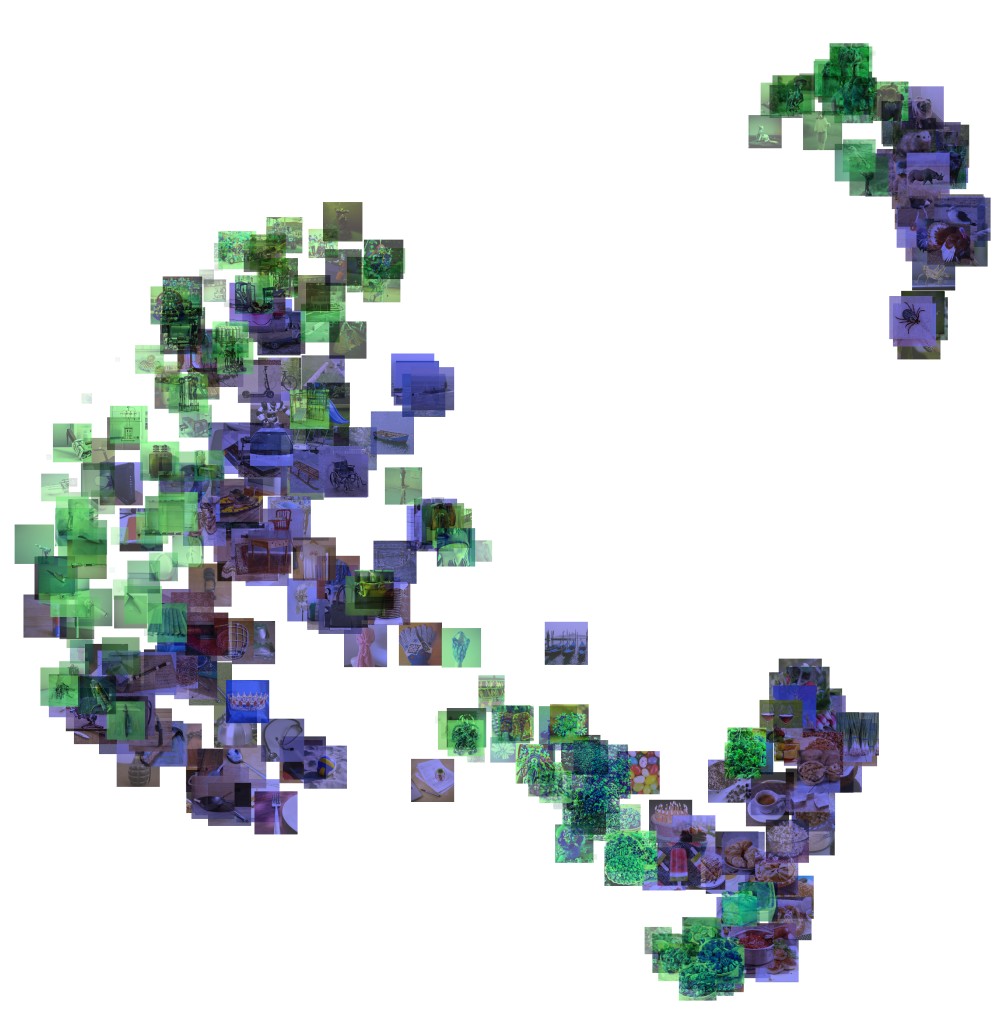

Figure A.2: UMAP CLIP alignment plot for test images for Subject 1. The blue-shaded images are encoded from the test images; the green-shaded images are encoded from the corresponding reconstructed images, with their size and opacity indicating the correlation of the CLIP vector between the the corresponding ground truth and reconstructed images. The ground truth images contain two clusters of images separated from the rest representing animals (top right) and food (bottom right), which reflect the 2 most prominent clusters in the reconstructed images as well

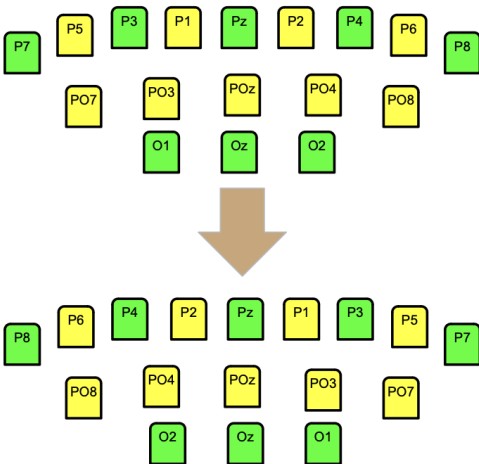

Figure A.3: Illustration of the EEG channel mirroring. In the mirroring experiment, the channels above the arrow are mapped to the channels below the arrow at test time (flipped around the midline). The electrodes in green are the ones used in the "half" experiment where half the electrodes are used.

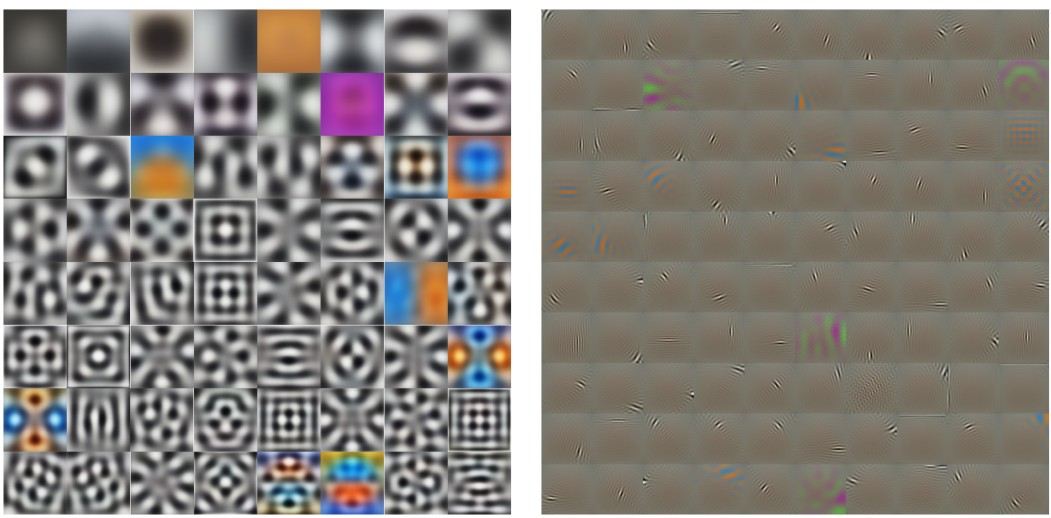

(a) Top 64 PCA Eigenvectors of ImageNet-64 by variance explained.

(b) Random ICA components of ImageNet-64

Figure A.4

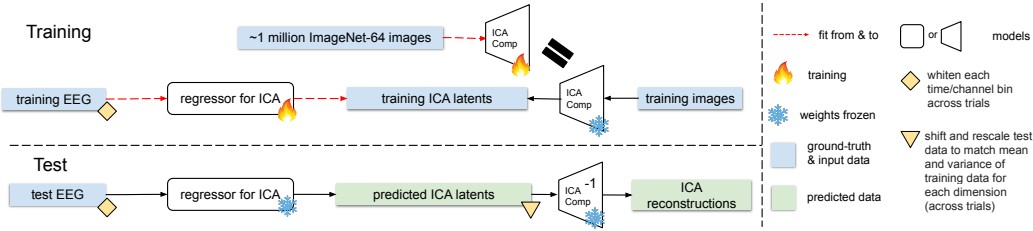

Figure A.5: Flowchart illustrating the ICA reconstruction pipeline.

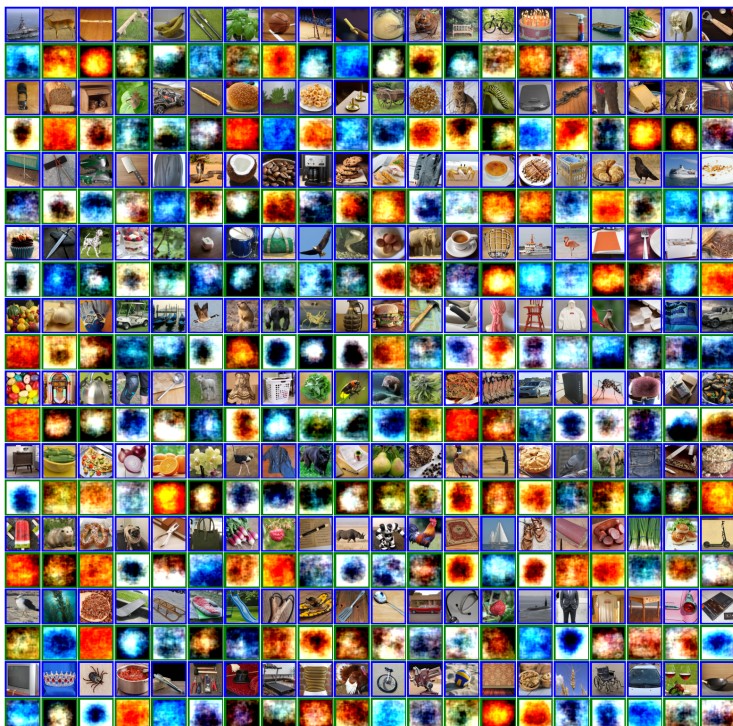

Figure A.6: Full PCA reconstructions for Subject 1.

Figure A.7: Full ICA reconstructions for Subject 1.

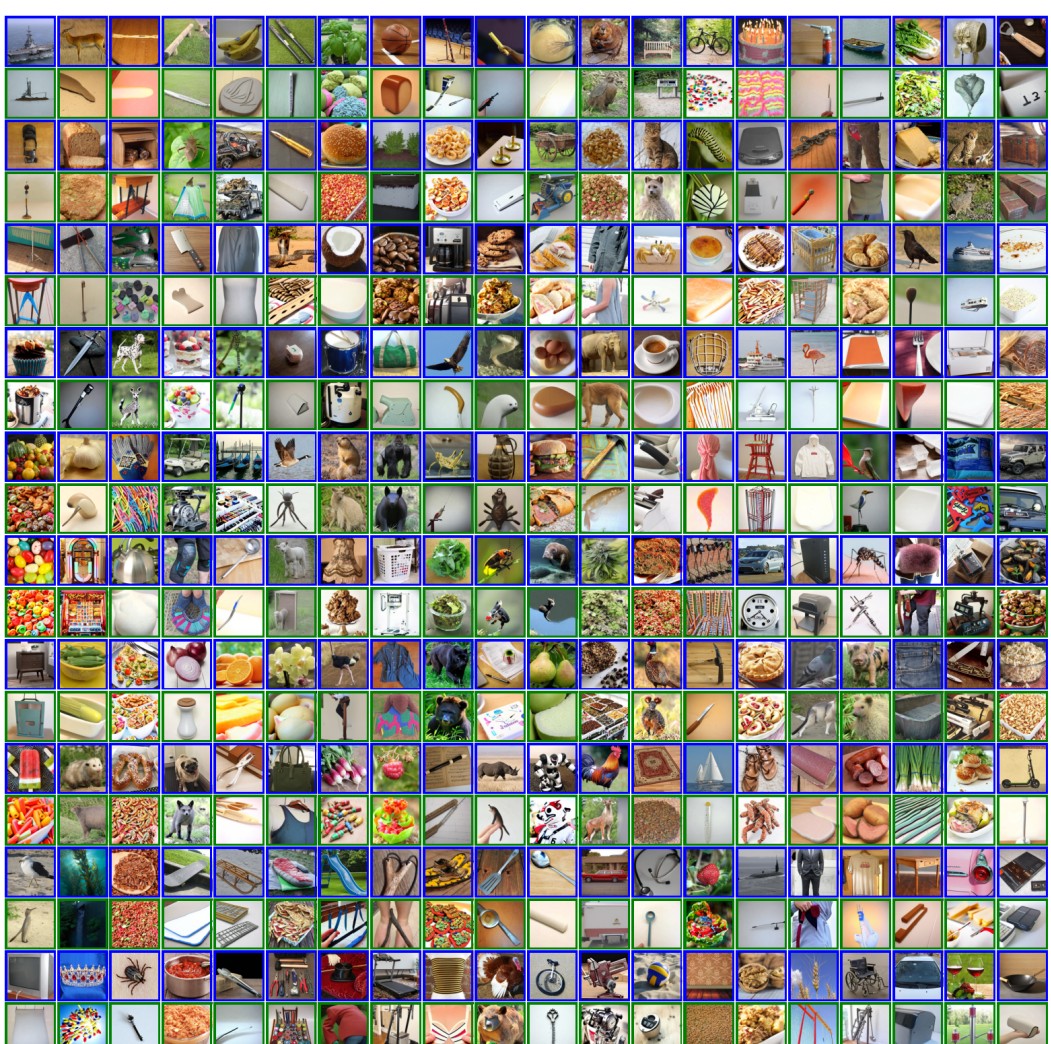

Figure A.8: Full reconstructions (Subject 1) using the unCLIP reconstruction pipeline.

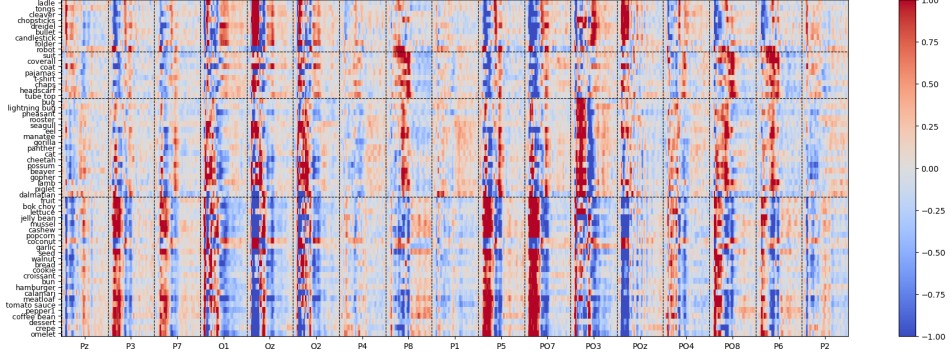

(a) EEG Patterns of CLIP (Subject 1). The hierarchical clustering on the CLIP embeddings extracted from the test images neatly organizes the 200 test categories into 3 general semantic groups (others, animals, food). Within the "others" group, it can be further subdivided into "small tools" and "clothing" with enough samples to see the pattern.

Figure A.9

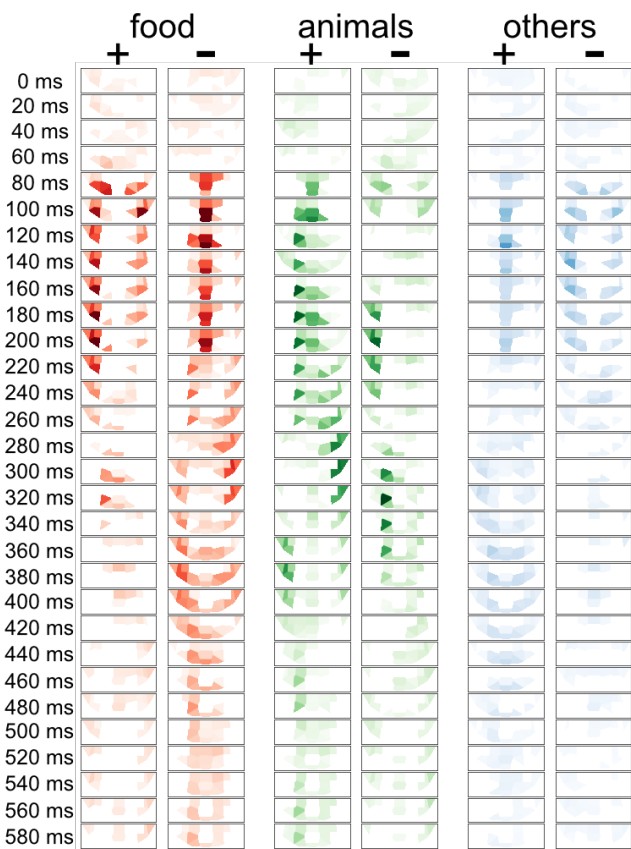

Figure A.10: The EEG spatiotemporal semantic map of Subject 1. A "+" indicates that category if there is a more positive voltage there; a "-" indicates that category if there is a more negative voltage there. Going from top to bottom, the spatial location sometimes "bounces" between left and right. This is in contrast to the cross-subject average version of this map. For individuals, there can be unique spatiotemporal features.

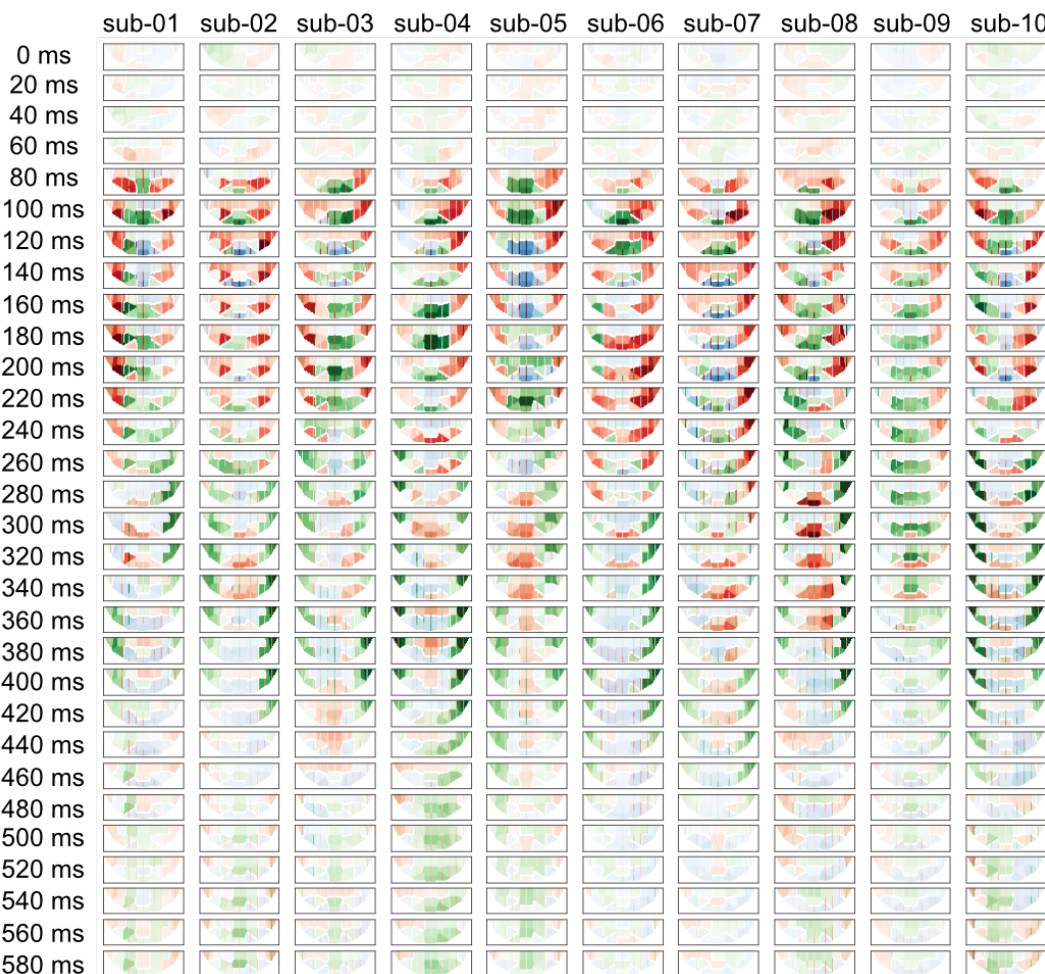

Figure A.11: The spatiotemporal semantic maps across 10 subjects. Based on Fig. A.12a, we take the average across images in the same general category, and plot the patterns of the channels in topological maps across time. Each miniplot represents activity over the back (visual) part of the brain for different subjects (columns) and different time segments (rows).Each column represents 1 of the 10 subjects. Each row from top to bottom represents a 20ms time window going from 0 to 800 milliseconds. We color the animal average green, food red, and everything else blue. With this plot, we can see the most general dynamical patterns for 3 general categories across 10 different subjects and can easily spot some consistent general trends with some individual differences. For each channel of a given time window, only the top positive category is shown as solid color, and bottom negative category is shown as vertical stripe. With this plot, we can see the most general dynamical patterns for 3 general categories. Note that PO8 and PO7 respond strongly for food-related pictures at around 100ms–this corresponds to the PO7 and PO8 activity in Fig. A.12a. Meanwhile POz and Oz respond strongly negatively for food-related pictures in the same time period.

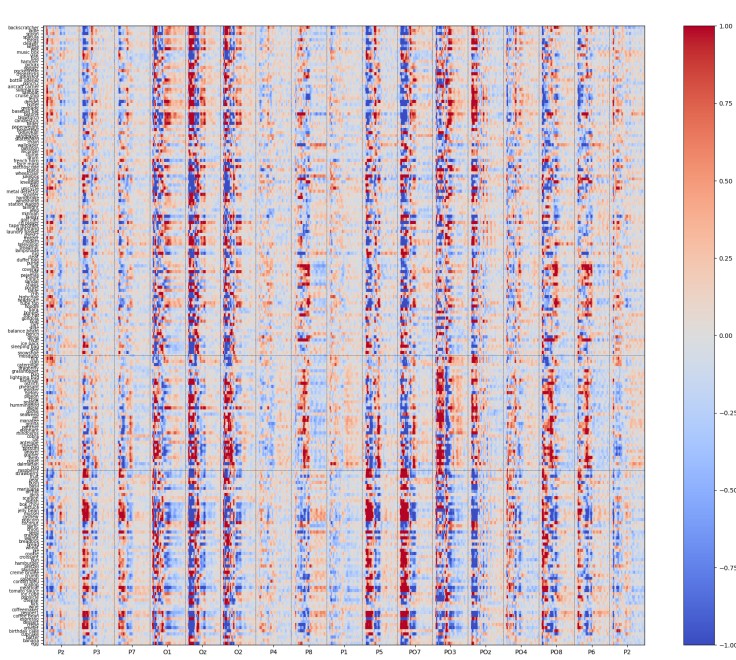

(a) The full "EEG pattern" of a single subject. Each row represents the EEG pattern of one of the 200 test images; the 2 horizontal dashed lines divide them into 3 general categories: food at the bottom, animals in the middle, and everything else at the top. The precise ordering was determined by hierarchical clustering of the CLIP representations of the images (not using EEG activity). Each column (between vertical black lines) represents an EEG channel; within each column, the smaller columns going from left to right are the time bins going from 0 to 800ms. Note the consistency in the patterns within the food and animal categories reflecting similar brain activity underlying perception of these objects.

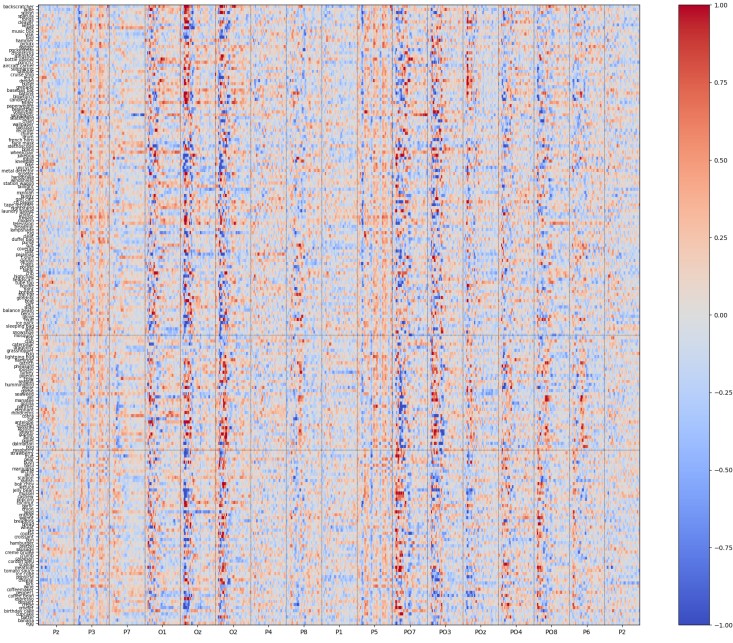

(b) The real EEG of the same subject. Note that the differentiating features look less pronounced.

Figure A.12

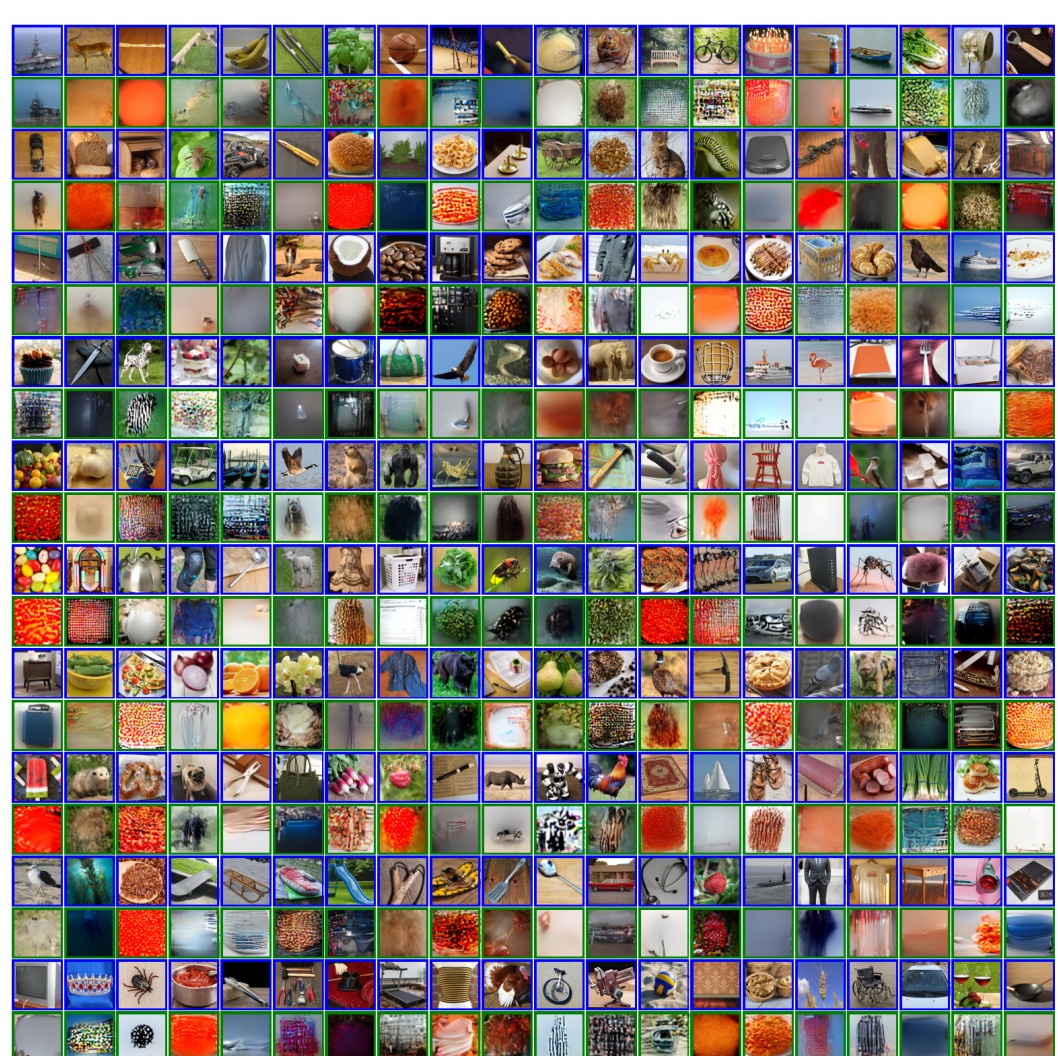

Figure A.13: VDVAE reconstructions (Subject 1)

## B    EEG WHITENING AND RE-NORMALIZATION OF THE PREDICTIONS

Before fitting, the EEG data is whitened to normalize each of the 1360 EEG dimensions across the 16540 training classes. The whitening process is expressed mathematically as:

$$xw_{ij} = \frac{x_{ij} - \mu_{ij}}{\sigma_{ij}}$$

where $xw_{ij}$ represents the whitened data, and $i$ and $j$ index the electrodes and time bins, respectively. The terms $\mu_{ij} = \text{mean}(x_{ij})$ and $\sigma_{ij} = \text{std}(x_{ij})$ denote the mean and standard deviation of $x_{ij}$, calculated across the training classes.

Once the predicted embeddings for all 200 test images are obtained, each dimension of the predicted embedding is normalized across all 200 predictions. Next, it is projected onto the mean and scaled to the standard deviation of the corresponding dimension from the training embeddings. This process is mathematically expressed as:

$$e_i = \left( \frac{e_i - \text{mean}(e_i)}{\text{std}(e_i)} \right) \cdot \text{std}(e_i^{\text{train}}) + \text{mean}(e_i^{\text{train}})$$

where $e_i$ represents the predicted embedding dimension, $\text{mean}(e_i)$ and $\text{std}(e_i)$ denote the mean and standard deviation of the $i$-th dimension across the test predictions, and $\text{mean}(e_i^{\text{train}})$ and $\text{std}(e_i^{\text{train}})$ correspond to the mean and standard deviation of the $i$-th dimension across the training embeddings.

