# S1 A LINEARITY ARGUMENT FOR THE EEG-TO-CLIP ALIGNMENT PROBLEM

## S1.1 MANIFOLD ISOMORPHISM

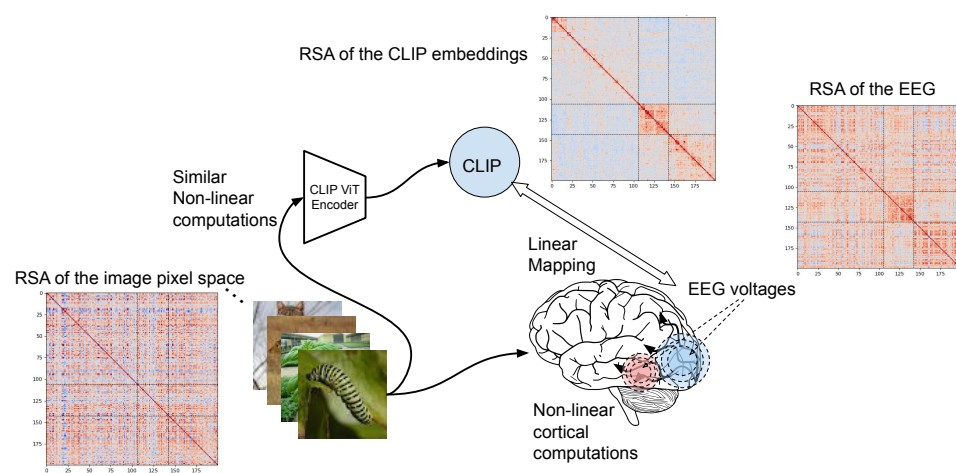

Figure S1

All neural decoding studies face the issue of manifold alignment; how to transform the representational space of the brain's activations to a human meaningful representation. The current problem involves two complex manifolds — CLIP space and EEG space — both generated by separate nonlinear processes (Figure S1). CLIP-image embeddings are generated by a large visual transformer (ViT-L/14), while the EEG activity from visual perception is generated by neural processing of visual stimuli summed with other internal and noise processes. While some studies have tackled decoding with neural networks or other non-linear algorithms, others have found comparable success (with explainability and computational efficiency benefits) using simple linear regression to map from brain data to target space. [see: Shen et al. (2019); Miyawaki et al. (2008) for examples in fMRI, see: Benchetrit et al. (2024) for an example in MEG] While the success of a linear mapping may seem surprising, we argue that its success depends on alignment of the CLIP and EEG representations.

It is clear that the mapping from pixel space to semantic representations is highly non-linear. This problem is solved in the brain in the mapping along the temporal visual pathway (Yamins et al., 2013). Similarly this mapping is fairly well solved in modern hierarchical computer vision systems and there is much interest in better understanding the relationships between these CV systems and the brain (Tu et al., 2018; Kong et al., 2020; Yamins et al., 2013) (though these types of findings aren't without some caution (Feghhi et al., 2024)).

EEG recording by nature of its summation of large-scale neural activity, is able to capture information from along the visual pathway including low-level pixel-related features as well as high-level semantic related features; these are all present in the EEG signal. Likewise the CLIP representation consists of representations of units from the CLIP hierarchical neural network, a large visual transformer (ViT-L/14) (Radford et al., 2021). The more similarly aligned the two representations are, the better a linear mapping can be learned between them. Linear maps (called linear probing) have been learned from CLIP representations successfully for many visual tasks (Radford et al., 2021).

We further answer this question in the context of this study by way of representational similarity analysis (RSA). Originally proposed to align neural data with computational model representations of image stimuli, RSA construes within-space correlation or similarity as a function of some property (e.g., stimulus) (Kriegeskorte et al., 2008). Here, we investigate RSA in source (EEG), target (CLIP), and stimulus (image) space to explain why a linear technique can solve the problem of manifold alignment. We consider the RSA of the EEG voltages, CLIP embeddings, and raw pixels for each of the 200 classes (Figure SF1.1 bottom left). The RSA plots of the EEG and CLIP embeddings

show a clear organization by class — animals, food, and other. The pixels themselves appear to be structureless in this regime, suggesting that EEG and CLIP perform similar non-linear transformations from pixel to semantic space. In other words, the two representational spaces are second-order isomorphic with respect to semantic class (not unlike the representational space based on shape similarity proposed by cognitive psychologists in the past — Edelman (2016); Choe (2019); Kriegeskorte et al. (2008). Such an isomorphism further motivates a simple linear projection between the manifolds, both of which exhibit sufficiently complex and functionally analogous computations. This also raises support for the validity of the semanticity of the textual representations with which CLIP is trained to align images.

## S1.2 NORMALIZATION

One thing that can impact the performance of the linear model is normalization. Our method has normalization at two different stages. First, the training and test EEG are whitened. The training and test sets contain different number of trials (4 for training and 80 for test), so when we average those trials, the resulting standard deviation is different:

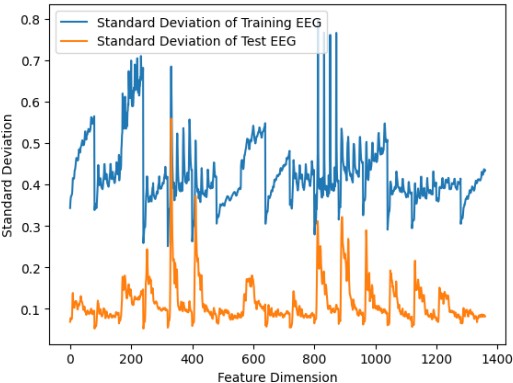

Figure S2

There are 1360 feature dimensions coming from 17 channels times 80 time points. The standard deviation of the training set is much higher than the test set due to having a lower number of trial averages. By whitening, it makes sure that the standard deviations of both the training and test sets are consistent with each other.

The second stage of normalization occurs post-prediction in the latent spaces. Here, the distribution of the test data is set to match the mean and standard deviation of the training data. This is because the latents predicted from whitened test EEG undershoot the standard deviation. Without the second stage of normalization, the data would under-span the latent space and produce suboptimal reconstructions (See Figure S3):

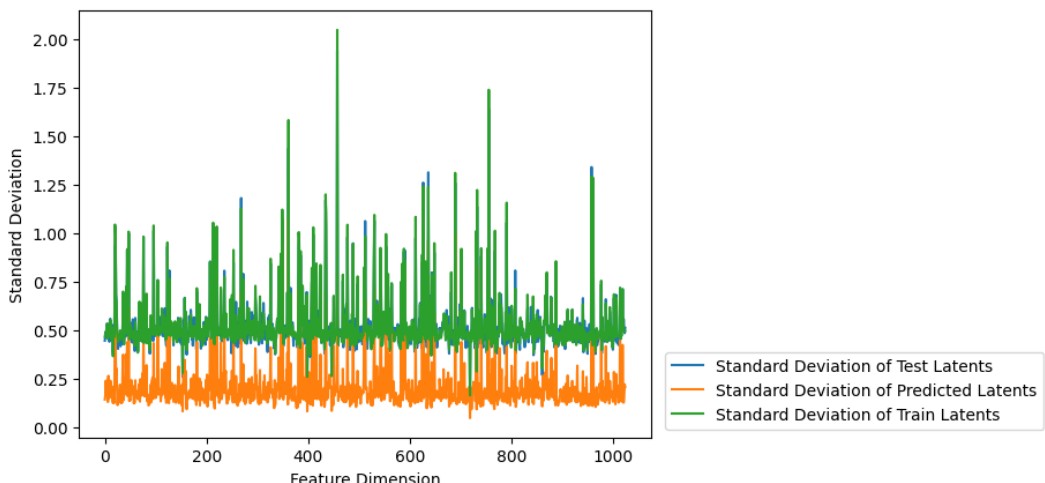

Figure S3

The second stage normalization projects the predicted latents to have the same standard deviation as the training latents. Since the training and test latents have almost the same standard deviation (See Figure S3), the predicted latents projected to the training standard deviation should have almost the same standard deviation as the test latents (See Figure S4):

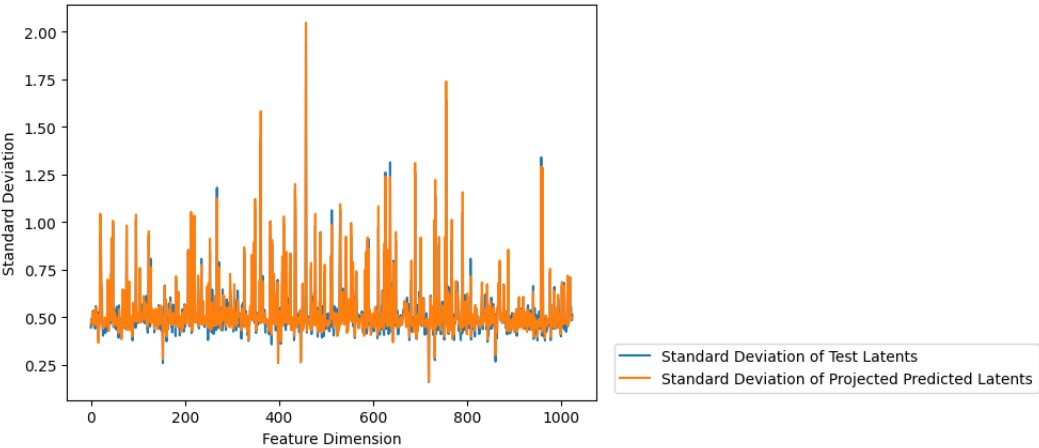

Figure S4

This is a simple but effective way of ensuring that the difference in the standard deviation due to mismatching trial numbers does not interfere with the reconstruction performance. In this example case from subject 1 using the unCLIP pipeline, the normalization raises the CLIP reconstruction performance from 0.780 to 0.807.

## S2  VISUAL FEATURE-RELATED EEG PATTERNS

Similar to the semantics, low-level visual features such as luminance, hue, and spatial frequency are also represented in EEG patterns. The decoding-encoding loop that we used to reveal features for semantics is general and can be applied to low-level visual features as well. Here we illustrate the EEG features relevant for low-level visual features such as luminance (brightness level), hue (color), and spatial frequency (smooth or "busy") embedded in the PCA, ICA and VDVAE spaces.

## S2.1 Luminance (Brightness)-Related EEG Patterns from PCA

We ordered the 200 PCA reconstructions by their luminance, increasing from left to right, and from top to bottom. Here we show an example ordering from Subject-1 (each subject is sorted by their own reconstructions):

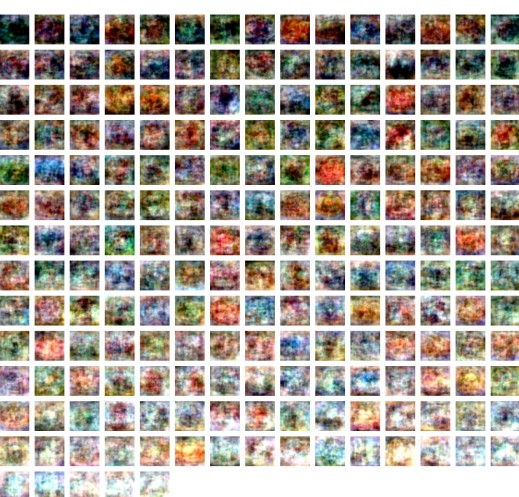

Figure S5

Here the reconstructions are sorted from darkest to lightest, regardless of color and other potential visual features. We applied the same technique that gave rise to the semantic map, but with PCA embeddings, to create EEG patterns linked to PCA instead (from Subject-1). The patterns are ordered by increasing luminance from top to bottom (Figure S6a). We also have the whitened EEG in the same order on the right (Figure S6b). These plots follow the same format at Figure A9.a and A12.a,b in the Main Document. We make them small here so you can see them together, but note that the electrode/time locations along the bottom and the ordering of the classes is the same as Figure A9.a and A12.a,b.

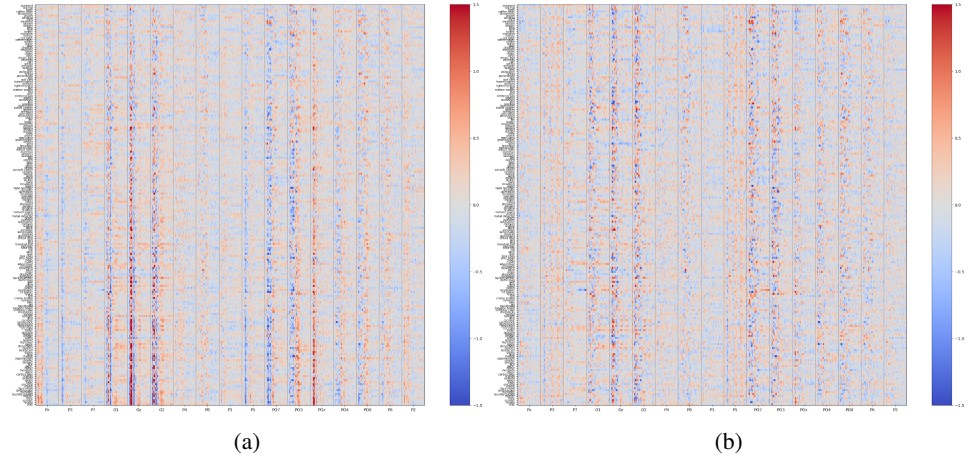

(a)             (b)

Figure S6

The PCA luminance-related patterns and the whitened EEG look quite different. The trend of going from negative to positive for O1, Oz, and O2 is still there, just to a lesser extent. This could be due to PCA encoding the general color as well. We show in the next section that the ICA color pattern is much closer to the whitened EEG.

### S2.1.1  DIFFERENCE MAP

To be able to visualize this continuous trend in a topographic map, we bracket the top and bottom 70 images and take the average of each group to form the "dark" map (corresponding to the reconstructions with the lowest luminance) and "bright" map. We subtract the dark from the bright map to form the "Difference Map":

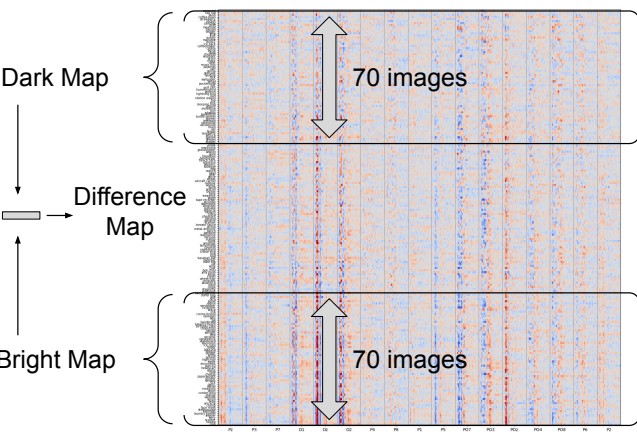

Figure S7

This allows us to show the difference map from all 10 subjects:

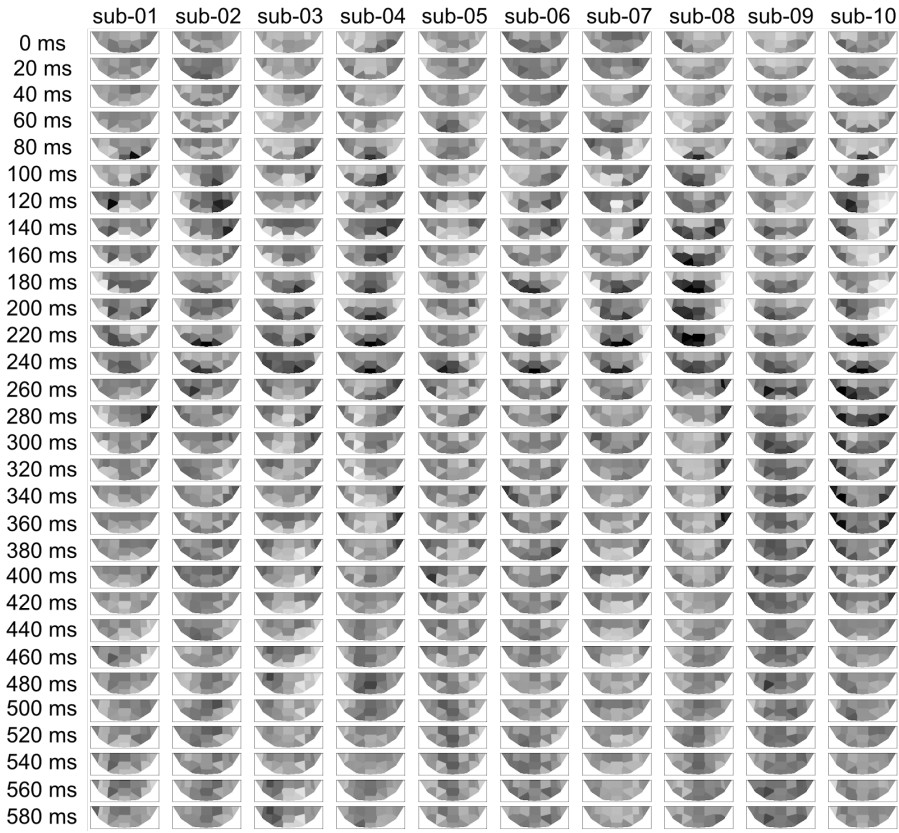

Figure S8

Figure S8 is a bright-minus-dark difference map of the PCA-linked EEG patterns. The brighter color for the electrode means more positive voltage. And since the opposite condition "dark" has generally voltage in the opposite direction than "bright", you can also think of the bright color as literally indicating high luminance (if there was a positive voltage there). Similarly, darker color technically means more negative voltage, but you can think of it as literally indicating lower luminance (if there was a positive voltage there). Medium gray here means indicating neither bright nor dark.

See Figure S15 for the cross-subject general trend.

## S2.2   HUE (COLOR)-RELATED EEG PATTERNS FROM ICA

We ordered the 200 ICA reconstructions by performing hierarchical clustering on them, which nicely organizes from warm to cold in terms of their hue (note that the warm to cold is not explicitly defined). Here is an example ordering from Subject-1:

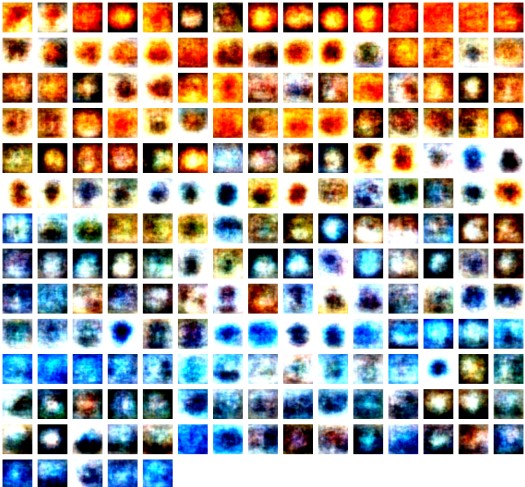

Figure S9

Here are the EEG patterns linked to ICA (Figure S10a) and the whitened EEG (Figure S10b) in this order (from Subject-1). Here, the EEG pattern and the whitened EEG look more similar to each other:

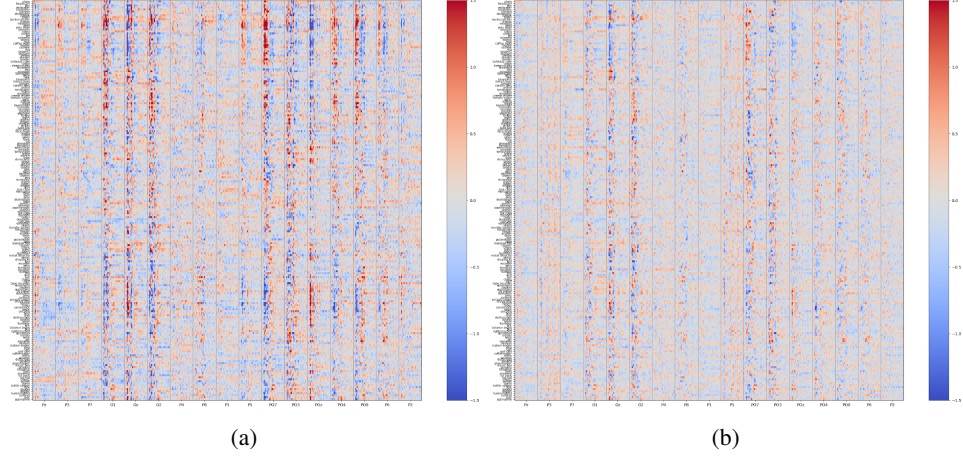

(a)                                          (b)

Figure S10

Here is the warm-minus-cold difference map from the top and bottom 70 reconstructions across all 10 subjects:

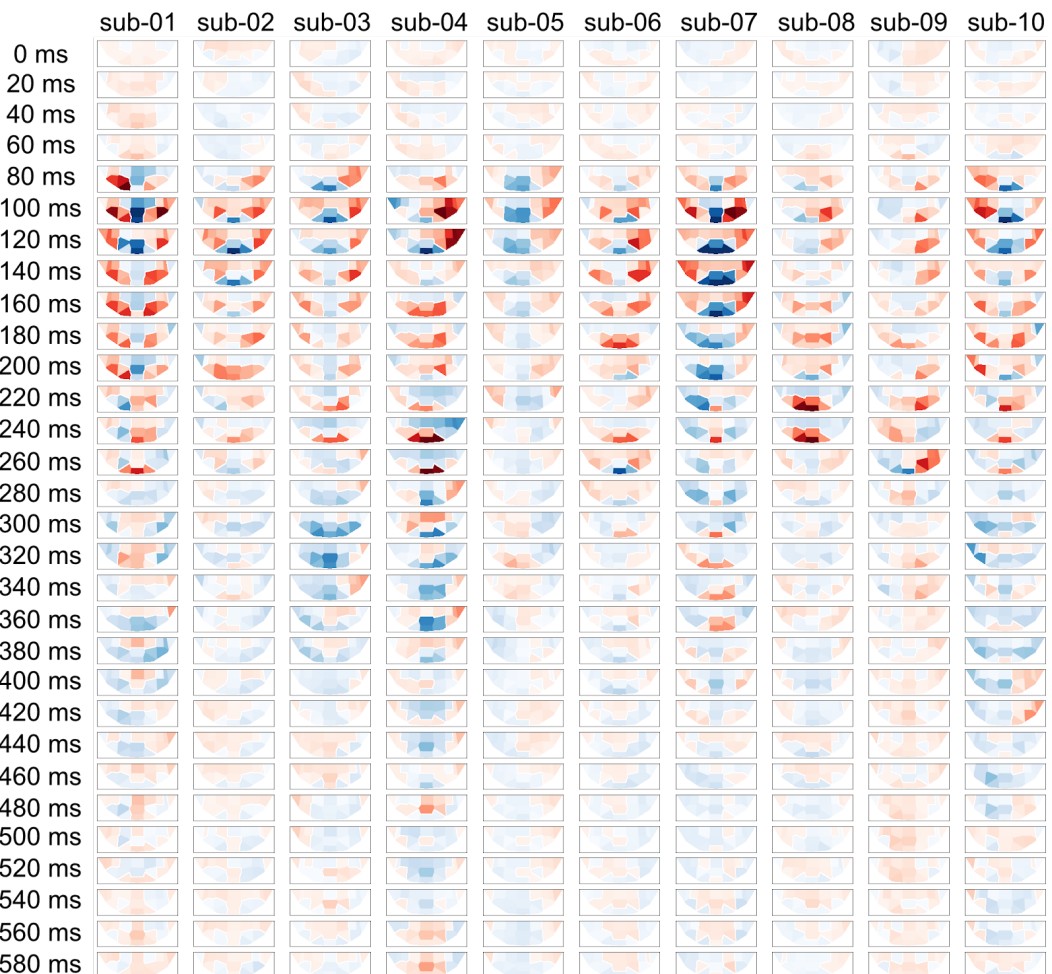

Figure S11

The red color indicates areas (electrode x time bins) where a more positive voltage indicates a warmer hue, and blue indicates where a more positive voltage indicates a cooler hue. The pattern is fairly consistent across subjects which is also reflected in Figure S15, which shows the cross-subject general trend.

## S2.3 TEXTUREDNESS (SMOOTH VS. TEXTURED)-RELATED EEG PATTERNS FROM VDVAE

The low-level reconstructions provided by VDVAE vary in texture: some images appear smooth while others appear highly textured. For example, in Figure 5 in the main paper, we showed the VDVAE reconstructions: the reconstructions for calamari and cashew look very textured and grainy since the original images look textured while the reconstructions of cheese wedge and CD-player are smooth.

If we sort the reconstructions from smooth to high-textured, they look like this:

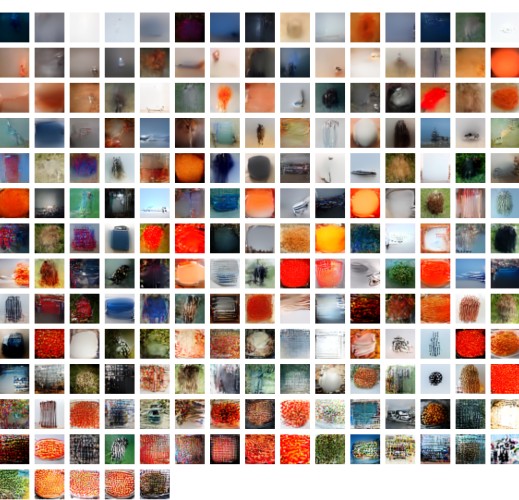

Figure S12

The reconstructions are sorted from smooth to textured regardless of the color and level of brightness. Here we plot the EEG patterns linked to VDVAE and the whitened EEG in the same order (from Subject-1):

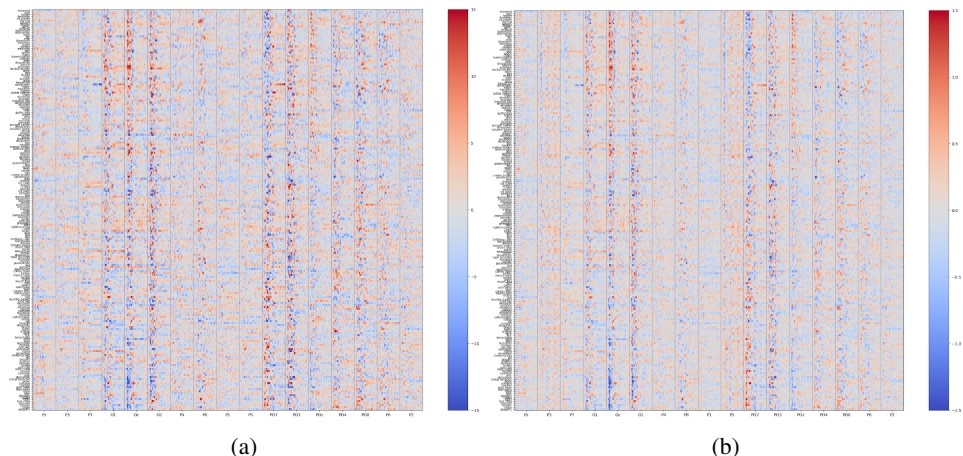

(a)                                        (b)

Figure S13

Here is the textured-minus-smooth difference map from the top and bottom 70 reconstructions:

Figure S14

There is no natural color code for textured or smooth, but Figure S11 uses red for positive and blue for negative, and it has red around the lateral electrodes and blue in the medial electrodes. To make this visualization theme consistent, Figure S14 uses orange instead of red and purple instead of blue, so the lateral still has a warmer color and the medial has a cooler one. The orange-ish color for the electrode means more positive voltage for higher textured stimuli. Similarly, purple-ish color indicates smoothness for more positive voltage.

See Figure S15 for the cross-subject general trend.

## S2.4 AVERAGE EEG PATTERNS SUMMARY

We take the average across the 10 subjects for EEG patterns linked to PCA, ICA, and VDVAE. Since we have the space here, we plot the two extremes separately rather than plotting just the difference between them:

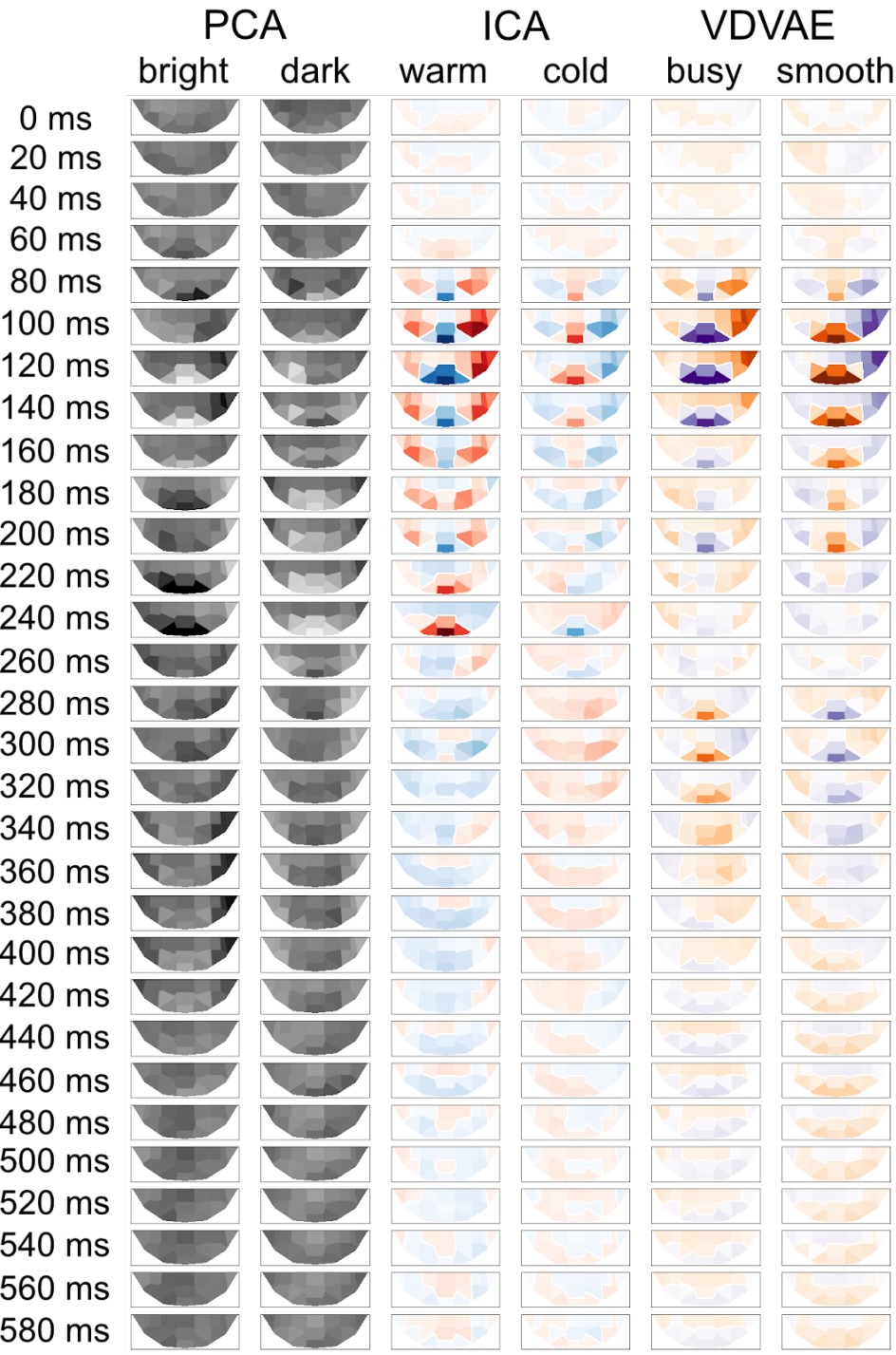

Figure S15

We see that the opposite features generally create EEG patterns with opposite voltages, this justifies taking the difference of the two extremes or just plotting one of them. Here we focus on just "bright", "warm", and "textured", and treat the negative voltage as the positive voltage from the opposite pattern.

For PCA, The strongest indication of high luminance occurs around 120 ms to 140 ms as you can see a small bright area around Oz. Then around 220 ms to 240 ms near Oz it reverses to indicating low luminance.

One previous study showed similar results with artificial dot stimuli (Sutterer et al., 2021):

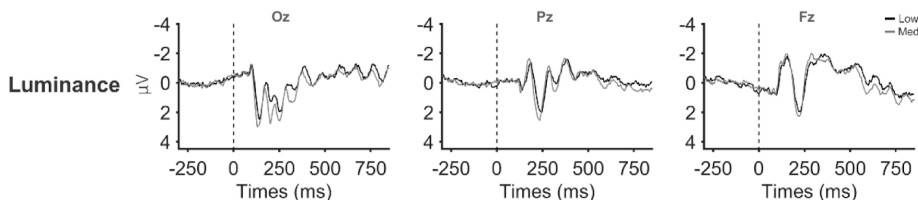

Figure S16: Adapted from Sutterer et al. (2021). Note that Oz and Pz are along the midline and overlap with the electrodes used in our study.

Their figure showed a positivity difference between medium and low luminance stimuli, particularly around Oz at around 120 ms, which is consistent with our results.

For ICA, we can see that the midline channels such as Oz and POz are more cold-tone biased around 80-140 ms while the more lateral electrodes are more warm-tone biased around the same time.

For VDVAE, we see again the medial versus lateral dichotomy: around 100 ms to 140 ms, medial electrodes have a bias for smoothness, and lateral ones have a bias for texturedness. But there are spatiotemporal nuances that make this map different from the hue-related difference map from ICA.

We are sure that there would be more low-level visual features to be found. But here we illustrate just 3 of the most easy-to-define ones.

## S2.5 RSA Alignment Evolution Across Different EEG Patterns

The EEG patterns emphasize the EEG features that the linear model finds useful, meaning those are the EEG features the linear model successfully maps to the target latent space. Therefore the EEG pattern reflects the target latent space: since the PCA latent space cannot be expected to encode high-level semantic information, the EEG patterns linked to PCA naturally de-emphasize all features related to semantics.

To illustrate the varying levels of alignment to the semantics, we perform RSA on the EEG patterns linked to PCA, ICA, and VDVAE latents, and compare them to the RSA of the whitened EEG and the RSA of test CLIP embeddings:

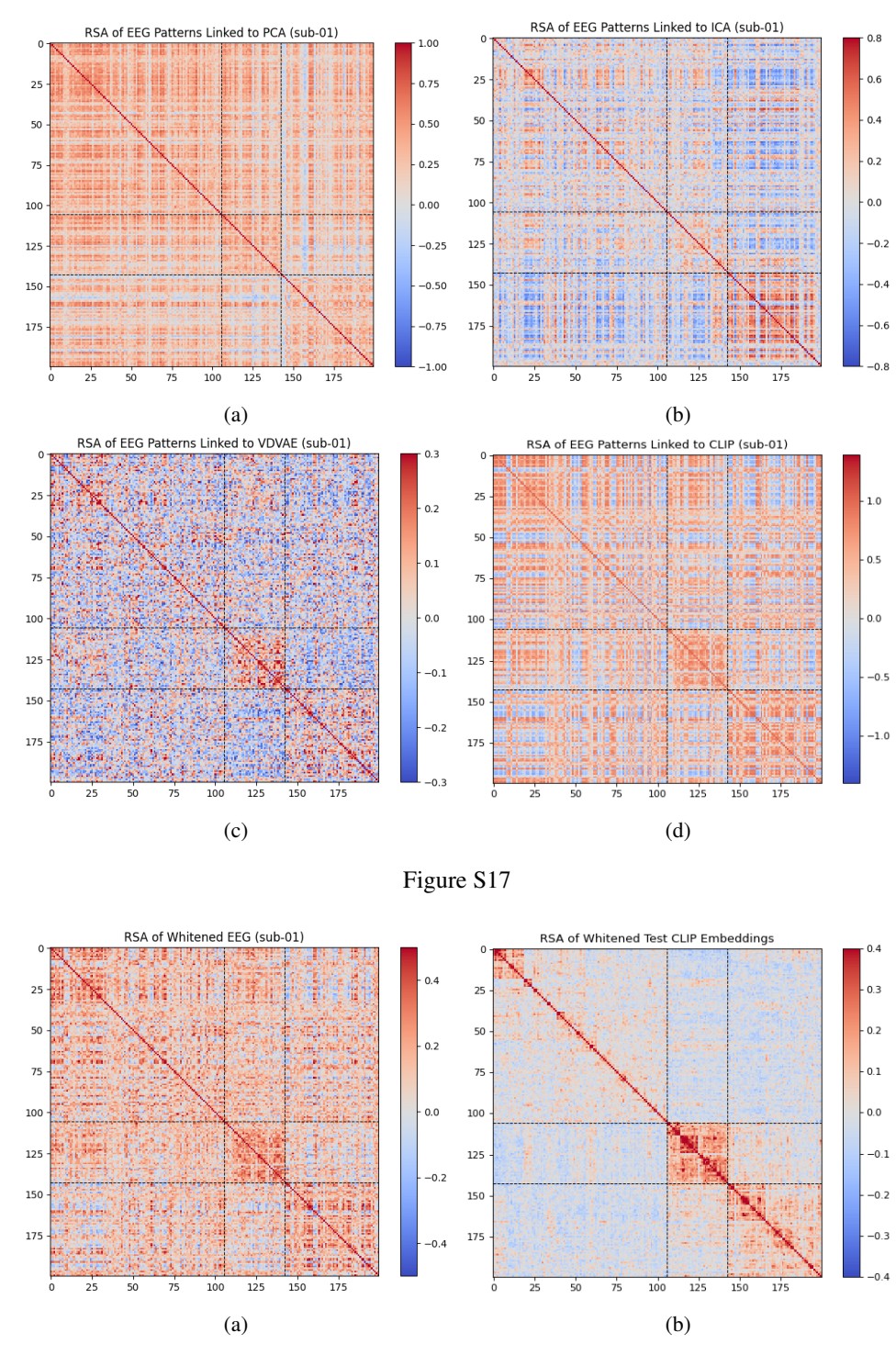

Figure S17

Figure S18

Figure S18a shows a similar structure as S18b. Figure S17a, S17b, S17c, and S17d show gradually increasing alignment to S18a and S18b.

S2.6   REPRESENTATIONAL AFFINITY DYNAMICS

Since the whitened EEG can be converted into EEG patterns linked to different latent spaces, the whitened EEG must contain the different types of feature-relevant information with reliable separability. How the EEG combines that information is of cognitive neuroscientific interest. Below are our preliminary results from investigating this question.

By performing RSA between whitened EEG and different EEG patterns, we can compute the structural distance between the 2 RSA's. The lower the distance, the more similar the whitened EEG is to the particular EEG pattern. We call this – the Representational Affinity – which measures the level of compatibility between 2 types of representations, e.g. whitened EEG and EEG pattern. Our hypothesis is that the whitened EEG combines different types of representations by following a low-level-to-high-level processing order in time. This requires us to perform RSA with a sliding time window (of 80 ms) on the whitened EEG and EEG pattern; we then compute the distance between the 2 RSA's across time – which we call the Representational Affinity dynamics. The key moment is when the distance reaches the minimum, which indicates EEG's moment of highest affinity to that particular type of representation. Here we show the cross-subject average EEG's Representational Affinity dynamics for PCA, ICA, VDVAE, and CLIP:

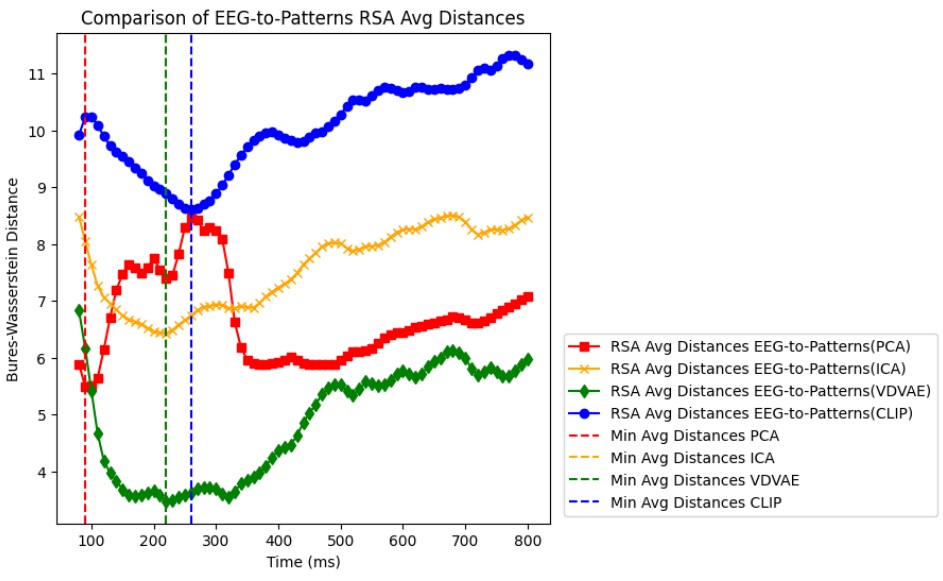

Figure S19

(The time on the x-axis means the right-end of the 80 ms time window.) Here we see a temporal sequence: PCA (low-level features) reaches the minimum early, followed by ICA and VDVAE (intermediate features), and CLIP (high-level features). Since we illustrated the association between PCA and luminance, between ICA and hue, and between CLIP and semantics, they follow a low-to-high level processing order. The idea that EEG moves from low-level to high-level processing over time is biologically plausible, given the known hierarchical visual processing pathways in the brain. The EEG has high affinity to PCA early on, but as the affinity to ICA and CLIP increases, the affinity to PCA drops. And as the affinity to ICA and CLIP decrease later on, the affinity to PCA rebounds back again. This suggests that the coding of low-level visual features does contradict with high-level in the EEG at some critical time range. However, this result is still preliminary and our claim about low-level to high-level processing requires further empirical support.

## S3  PCA Reconstructions with the Original Image Principal Components (without EEG)

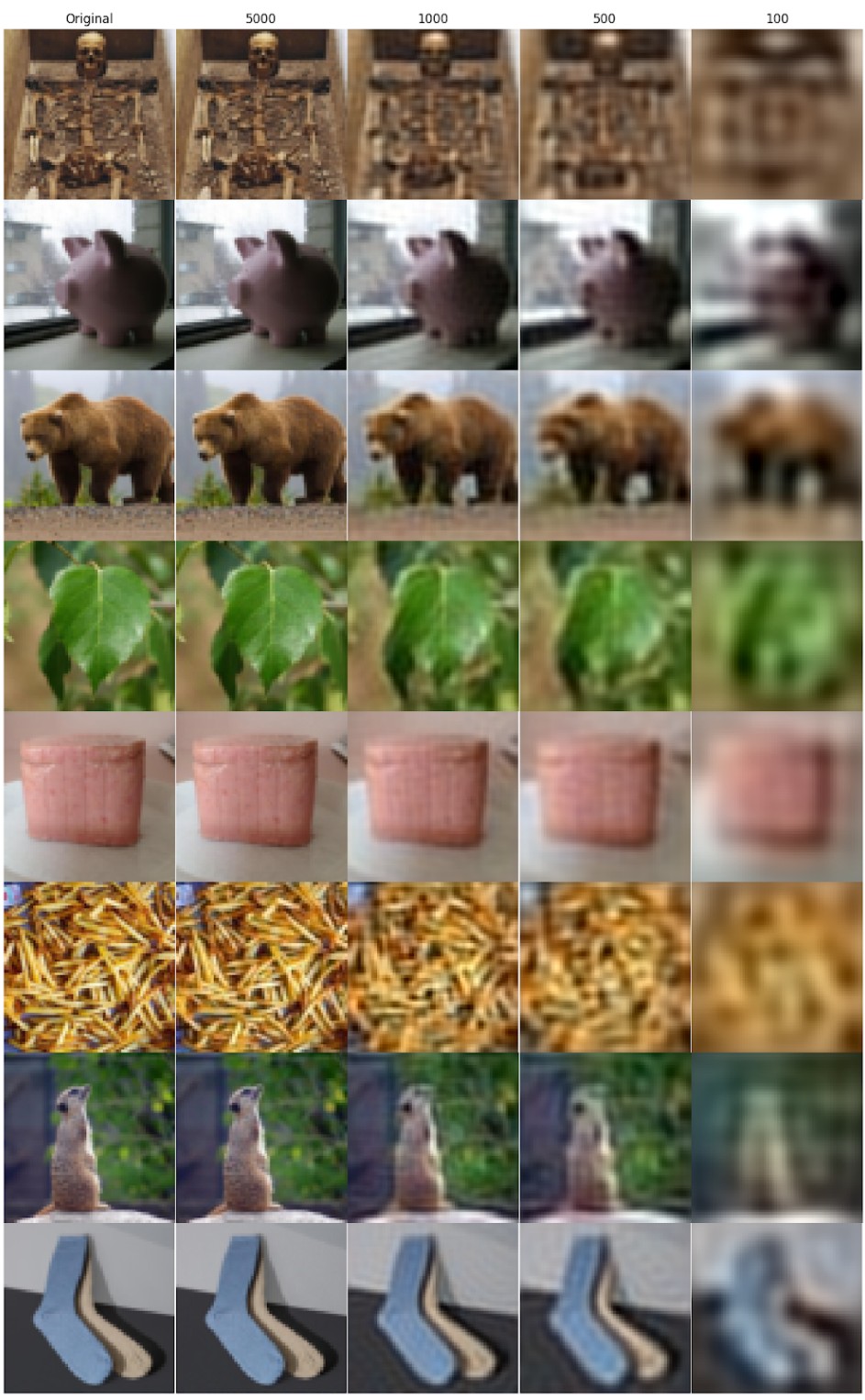

Figure S20

## S4 Transfer learning through data augmentation

To make the reconstruction pipeline more feasible for BCI, we explored the issue of transfering knowledge from training on one participant to future participants allowing for less training data collection.

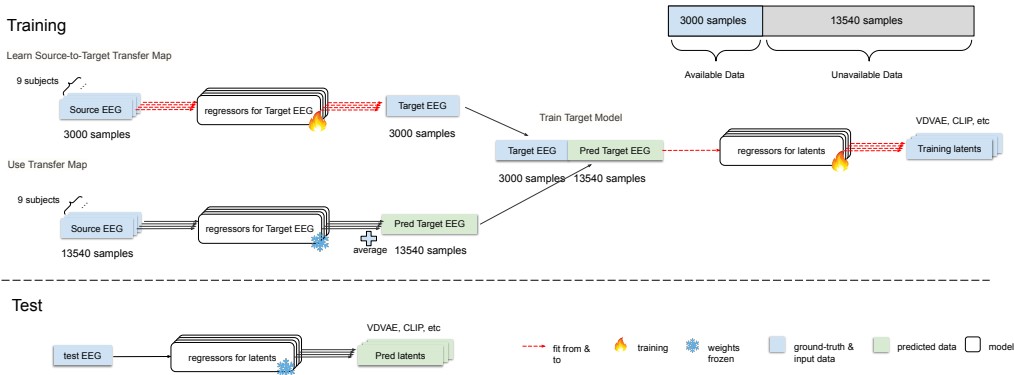

Figure S21: Flowchart illustrating the cross-subject reconstruction pipeline.

Here, we fit a regression model to a small subset of the Target subject's training data (i.e. EEG on a few training images) from EEG data from other (Source) subjects on those same images (i.e. we learn a mapping from $EEG_{\text{SourceSubject}} \rightarrow EEG_{\text{TargetSubject}}$). We then predict the EEG response to the remaining images for the Target subject from those learned mappings. (We average the predictions from each of the other Source subjects to form the augmented data for the Target subject.) A model is then trained with the (small amount of) Target subject training data and the predicted (mapped) Target subject data to the other training images. Finally the model is tested on test data from the Target subject.

For this study we used the full training data from subjects 2-10 and used a small amount of training data from Subject 1. Mappings were learned from each of the other subjects to Subject 1 on the small amount of training data. Regularization strength of 1000 was used for this mapping. The other training images were then passed through each of the Subject models and then through the corresponding transfer mapping to generate "augmented" data for training the regression mappings for subject 1. These data were averaged from each of the other 9 subjects before being used for training. Subject 1's test data was then used for testing the model. This method was compared to a model that used only Subject 1's small amount of training data (and was tested on the same test data). Note that for selecting the available training data, we used the alphabetical order of the training image names, which is not the most realistic assumption about training data sampling.

Results are shown in Figure S22a. The figures show increased performance with increased training size but at all training sizes there is an improvement with our augmented transfer method.

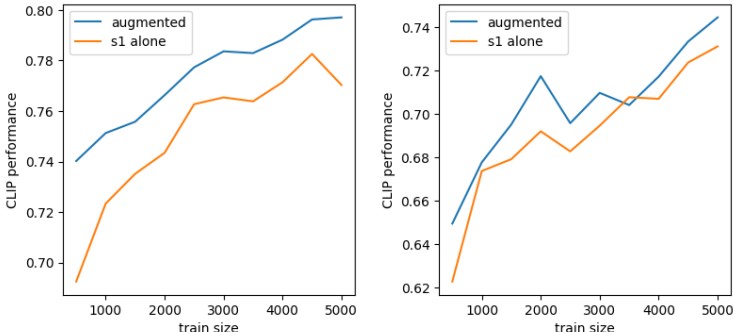

(a) CLIP performance for Subject 1.    (b) CLIP performance for Subject 1 with 10 trials average for the test EEG.

Figure S22: Transfer learning CLIP performance for Subject 1 with random seed 0. The orange curve (s1 alone) gives the performance using only the "train size" amount of training data from S1. The blue (augmented) curve shows the result using augmented data from the other subjects (using the method described in the text). No error bars are shown due to having only one subject with one random seed. a) shows the performance using the standard 80 test trials per image and b) shows performance using only 10 test trials per image (as a way of examining faster reconstructions).

## S5 ADDITIONAL FIGURES

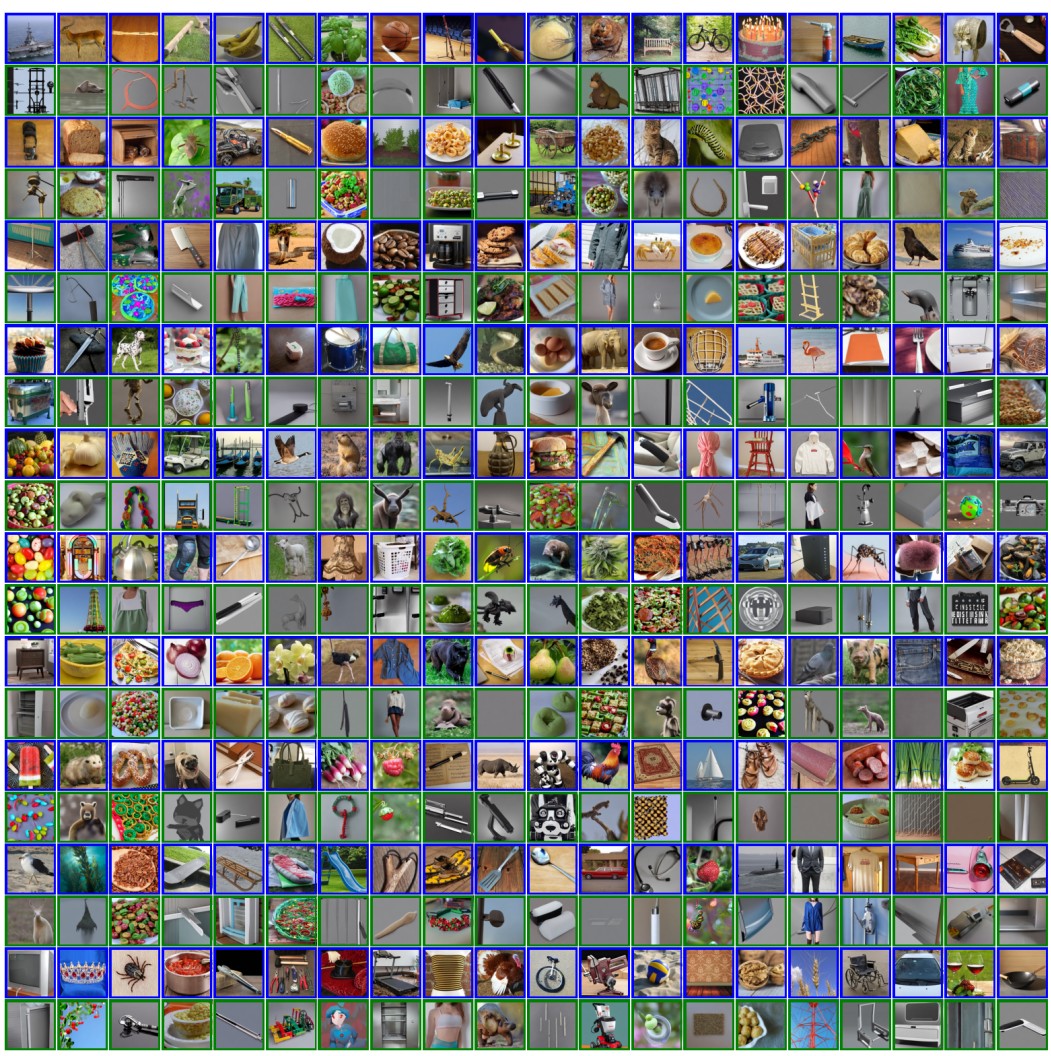

Figure S23: CLIP-Text only reconstructions for Subject 1 with random seed 0.

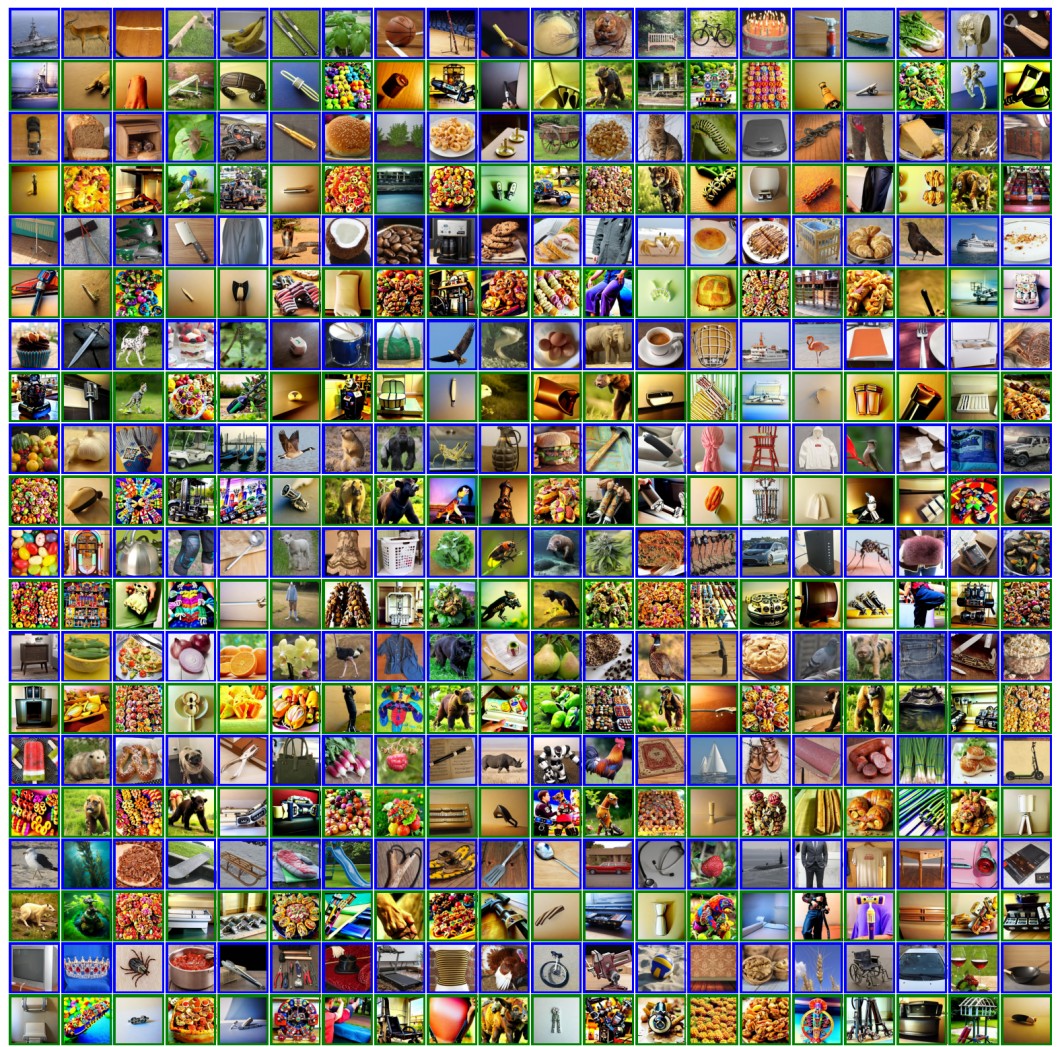

Figure S24: CLIP-Vision only reconstructions for Subject 1 with random seed 0.