# OpenReview forum: "Perceptogram: Visual Reconstruction from EEG Using Image Generative Models"
_ICLR.cc/2025/Conference — Submitted to ICLR 2025_

### Official Review · Reviewer_P37j · 2024-10-28

**Soundness:** 2
**Presentation:** 2
**Contribution:** 3
**Rating:** 5
**Confidence:** 4

**Summary:**

This paper highlights the method of generating images from EEG signals by CLIP embedding. In the visual representation and construction field, EEG is less frequently used and has better temporal resolution instead of spatial resolution, so it may be useful for future work related to brain-computer interface. The research regarding the scalp-asymmetry effect, time-swap effect, and spatiotemporal semantic map described some important internal mechanisms of visual reconstruction.

**Strengths:**

1. A simpler model that has state-of-the-art performance. Every stage of the whole model is highly modular. Multimodal extraction with different models. In 3.3 we can see that VDVAE-based reconstructions is useful to exctract the latent space from EEG signals.

2. The method of observing time-swap effect and spatiotemporal semantic map is fascinating, making full use of the characteristics of EEG's high time resolution. From the spatiotemporal semantic map (Figure 9), the trends show some consistent and universal dynamic patterns to help researchers understand the time-dependence reactions of the brain when dealing with these different categories.

**Weaknesses:**

1. The contribution of this paper to the community appears limited, as it does not provide new insights or innovative research directions. The repackaging of existing work—specifically the separation of CLIP embeddings into text and image components—does not signify substantial progress within the field.

2. The absence of a detailed model explanations, and mathematical rigor undermines the paper's contribution. Key sections, particularly between lines 130 and 150, are inadequately polished; there is a pressing need for more precise mathematical formulations and clearer textual explanations to enhance comprehension and rigor. And the presentation of the results is unpolished, e.g. fig6, text size should be unified, and in fig.7 texts should be aligned, etc.

3. In line 349, it mentioned the symmetry about the vertical midline, especially the images that have strong diagonal components, but this conclusion is based on a few specific examples (such as columns 1 and 11).

4. In line 354, the analysis mentioned that the "animal nature" of animal images sometimes disappeared under mirror conditions, but it is not clearly stated whether this is consistent with the difference between the hemisphere between the hemisphere.

5. The authors rely on a straightforward ridge regression model to align EEG embeddings with CLIP(Radford, et al., 2021), raising concerns about whether such a simplistic encoding structure can sufficiently capture the complex spatial and temporal dependencies inherent in EEG data. The reconstruction pipe used in this paper doesn't contain much of a trick, the framework failed to provide images with richer information to do some complex tasks. But it turned out to be much better than previous methods, and I doubt it.

6. The discussion of scalp-asymmetry effects is superficial, lacking the necessary depth and rigor. Furthermore, the placement of key figures, such as Figure 3(d), could be optimized for reader accessibility, as its current location disrupts the flow and readability of the analysis.

**Questions:**

1. I would like to see a broader range of examples demonstrating the types of reconstructions achieved by your pipeline. Could you provide additional examples or case studies that illustrate its applicability and effectiveness?

2. Asking author to justify the impressive EEG encoding performance achieved with such a simple ridge regression model? What underlying principles or characteristics of the EEG data allow this approach to succeed, despite its simplicity?

3. Can author elaborate on any theoretical insights or assumptions that guided the design of your model? A clearer connection between your method and the principle would strengthen your contributions.

---

> ### Author Response · Authors · 2024-11-24
> **Response to P37j**
>
> Thank you for your review of our manuscript. We have responded to your written comments below, with references to added supplementary materials for further clarification.
>
> **Response to Weakness 1:**
>
> While brain-to-image reconstruction studies are not new, most fail to address the question of representational mapping, relying on the black box of diffusion models. That is to say, our approach seeks to elucidate the underlying dynamics when mapping between image (pixel) space, embedding space, and EEG space. Not only do we show that a dimensionally-reduced, more efficient CLIP model – stable unCLIP – can be used for these pipelines, but we also present various features of the image space that are impacted by the encoding scheme (e.g., the representational affinity, Section S2) and the spatiotemporal impact of the EEG space on the reconstructed image space (e.g., the scalp asymmetry and time-swapping effects, Section 4).
>
> **Response to Weakness 2:**
>
>     “​​Key sections, particularly between lines 130 and 150, are inadequately polished… fig6, text size should be unified, and in fig.7 texts should be aligned, etc.”
> We have improved the flow of that section and updated the figures accordingly.
>
>     “there is a pressing need for more precise mathematical formulations”
> Our latent alignment relies on Ridge Regression, a simple yet robust method that incorporates regularization to prevent overfitting. It can be implemented through well-validated libraries (e.g., Scikit-Learn, Matlab). While it may lack the complexity of neural-network-based approaches, many of these methods also lack precise mathematical formulations, relying instead on empirical optimization. We agree that the whitening of EEG and the renormalization of the predictions can be better explained with mathematical formulations, which we now include in the Appendix Section B.
>
> **Response to Weakness 3:**
>
> We have subsequently conducted a more systematic test to quantify the orientation change due to the mirroring of the electrodes and the results were not statistically significant. We believe it may be due to the low number of asymmetric images in the test data, and thus a dedicated test with more asymmetric images may be required. We have taken off the paragraph about orientation mirroring.
>
> **Response to Weakness 4 and 6:**
>
> Yes, the hemispheric differences in the semantic maps of the ten subjects provide insight into why mirroring the electrodes can disrupt semantic information. While the cross-subject average semantic map shows "animalness" corresponding to more medial activations and "foodness" to more lateral activations, individual maps often reveal distinct differences and asymmetries. Observing semantic disruptions under mirror conditions motivated us to develop a method for linking specific EEG patterns to semantic features. This led to the creation of our decoding-encoding loop for semantic mapping.
>
> **Response to Weakness 5:**
>
> Please see our response to Question 2 below.
>
> **Response to Question 1:**
>
> We initially validated our model using the THINGS-MEG dataset but shifted our focus to THINGS-EEG 2 after seeing the feasibility of reconstructions with that dataset. Our next step is to revisit THINGS-MEG and adapt our decoding-encoding loop to explore whether MEG can produce a more spatially precise semantic map. In parallel, we plan to compute the semantic map for the NSD dataset and compare it with the auditory-derived Huth Semantic Map to evaluate cross-modal alignment.
> Additionally, we are extending visual reconstruction experiments by introducing variations in experimental design, which we believe are crucial for providing a more comprehensive range of examples.
>
> **Response to Question 2:**
>
> We address this question in Section 1 of our supplemental materials section, in which we first note that ridge regression from the brain to image latent space has been used before in reconstruction studies using fMRI data. Furthermore, the performance of the simple model can be explained by the overdetermination of the mapping problem, previous mapping studies of neural network layers to layers in the visual system, and our own RSA investigations. We discuss this in more detail in Section S1.1 in the Supplemental Material.

---

> > ### Comment · Reviewer_P37j · 2024-11-25
> >
> > 1."Not only do we show that a dimensionally-reduced, more efficient CLIP model – stable unCLIP – can be used for these pipelines"
> >
> > As far as I know, image reconstruction using CLIP embedding after dimensionality reduction or linear mapping is already relatively mature [1][2].
> >
> > 2."Our latent alignment relies on Ridge Regression, a simple yet robust method that incorporates regularization to prevent overfitting."
> >
> > The reason why ridge regression is powerful in modeling fMRI is that the features of the original BOLD have been extracted when preprocessing to obtain voxels. From the technology of neural engineering perspective, I am not convinced that ridge regression still has a strong role to play in EEG modeling.
> >
> > References
> >
> > [1] Oikarinen T, Weng T W. CLIP-Dissect: Automatic Description of Neuron Representations in Deep Vision Networks[C]//The Eleventh International Conference on Learning Representations.
> >
> > [2] Wei C, Zou J, Heinke D, et al. CoCoG: Controllable Visual Stimuli Generation based on Human Concept Representations[J]. arXiv preprint arXiv:2404.16482, 2024.

---

> ### Author Response · Authors · 2024-11-24
> **Response to P37j (Part 2)**
>
> **Response to Question 3:**
>
> The base design of our model—mapping neural activity to a latent space and subsequently leveraging a diffusion model for image generation—builds on methodologies used in prior studies. While previous works have also explored encoding images into EEG, these approaches often treat encoding and decoding as separate processes, without explicitly identifying the EEG features utilized by decoding models.
>
> Our key methodological innovation lies in the introduction of the decoding-encoding loop, which establishes a direct link between model predictions and specific EEG patterns. These patterns provide a tangible visual reference for guiding future experimental variations and improving interpretability.
>
> In addition to the model design, we contribute theoretical insights by investigating the relationship between latent space representations (e.g., ICA, VDVAE) and image features (e.g., color, texture). This connection deepens our understanding of how neural activity corresponds to visual perception, offering a foundation for both methodological and theoretical advancements.

---

> > ### Comment · Reviewer_P37j · 2024-11-25
> >
> > "Our key methodological innovation lies in the introduction of the decoding-encoding loop, which establishes a direct link between model predictions and specific EEG patterns."
> >
> > I don't quite understand what you mean by the "decoding-encoding loop". As far as I know, this paper does not design a closed-loop iterative process.
> >
> > I think some of the author's analyses are interesting but need time to refine, and the technical contribution of the paper as a whole is not outstanding. Therefore, I suggest that the paper may need going through another round of review to get a fairer judgement and will keep my decision.

---

> > > ### Author Response · Authors · 2024-12-04
> > > **Response to P37j Comment 2 and Comment 3**
> > >
> > > Often linear models work comparably to nonlinear methods due to the noisiness of the EEG [1]. The whole Cognitive Neuroscience ERP field is based on linear analysis (averaging) of EEG signals.
> > >
> > > While there are differences in the temporal and spatial characteristics of fMRI and EEG, they can both achieve reasonable decoding with sufficient averaging to increase the signal-to-noise ratio (SNR). For fMRI, preprocessing often involves general linear models (GLMs) (to estimate beta values), which effectively averages trials while avoiding temporal smearing due to the hemodynamic response delay. This averaging is really the driving force of the SNR.  Like fMRI,
> > > time-locked averaging of EEG is a common pre-processing for EEG (and forms the basis for ERP analysis).  The NSD has 3 trials per image, and THINGS-EEG 2 has 80 trials per test image reflecting the noisier nature of EEG, but with the 80 trials, similar results to fMRI with fewer trials can be achieved. (Note also the fMRI trials last longer, so “amount of time” is closer than appears from trial numbers).
> > >
> > > We feel that our work may be being held to a higher standard than methods that employ complex neural network architectures.  Those methods  have not demonstrated superior performance compared to ours, but appear to count as technical contributions just because they used a novel network architecture.  The complex network modeling works did not provide rigorous mathematical justifications for why their nonlinear decoders are effective. They did not reveal any specific EEG features their models extract. Our approach, using simple ridge regression, offers both interpretability and performance without relying on unnecessary complexity.
> > >
> > > The decoding-encoding loop is part of our contribution and is motivated by the common practice of visualizing common spatial patterns [2] in EEG and best practices for understanding linear models [3].
> > > In our decoding-encoding loop we:
> > > 1) Take the EEG for a given image and use the decoder to get the predicted latent embedding vector
> > > 2) Use the “encoding” linear network to project that predicted latent back to the EEG space
> > > 3) Visualize that EEG pattern
> > >
> > > The newly encoded EEG (the EEG pattern) has a direct and detailed linear relation with the decoded prediction: for instance, a unit increase in the “animal-related” EEG pattern corresponds to a unit increase in the “animalness” dimension of the predicted embedding.
> > > The EEG pattern reflects the relevant features in the EEG that were used for decoding into the latent space.   (Notably the patterns will also depend on the latent space used;  different latents will reveal different aspects of the EEG.)  This explicit linkage enhances interpretability and provides a precise framework for identifying meaningful EEG patterns that our model has extracted that can serve as a reference for more advanced models.
> > >
> > > References
> > >
> > > [1] Schirrmeister, R. T., Springenberg, J. T., Fiederer, L. D. J., Glasstetter, M., Eggensperger, K., Tangermann, M., Hutter, F., Burgard, W., & Ball, T. (2017). Deep learning with convolutional neural networks for EEG decoding and visualization. In Human Brain Mapping (Vol. 38, Issue 11, pp. 5391–5420). Wiley. https://doi.org/10.1002/hbm.23730
> > >
> > > [2] Blankertz, B., Tomioka, R., Lemm, S., Kawanabe, M., & Muller, K. (2008). Optimizing Spatial filters for Robust EEG Single-Trial Analysis. In IEEE Signal Processing Magazine (Vol. 25, Issue 1, pp. 41–56). Institute of Electrical and Electronics Engineers (IEEE). https://doi.org/10.1109/msp.2008.4408441
> > >
> > > [3] Haufe, S., Meinecke, F., Görgen, K., Dähne, S., Haynes, J.-D., Blankertz, B., & Bießmann, F. (2014). On the interpretation of weight vectors of linear models in multivariate neuroimaging. In NeuroImage (Vol. 87, pp. 96–110). Elsevier BV. https://doi.org/10.1016/j.neuroimage.2013.10.067

---

### Official Review · Reviewer_qxem · 2024-10-31

**Soundness:** 3
**Presentation:** 3
**Contribution:** 2
**Rating:** 5
**Confidence:** 5

**Summary:**

The paper presents an approach for reconstructing viewed images from EEG recordings. The authors employ a linear decoder that maps EEG signals to image latents, leveraging CLIP embeddings and latent diffusion models to achieve state-of-the-art quantitative reconstruction performance. The study explores the use of PCA and ICA components to capture luminance and hue-related information from EEG. The linear model provides interpretable EEG features for differentiating semantic categories of images and creates spatiotemporal semantic maps reflecting the temporal evolution of class-relevant semantic information.

**Strengths:**

The paper introduces an application of EEG in the domain of image reconstruction, which is traditionally dominated by fMRI and MEG. The use of CLIP embeddings and latent diffusion models for EEG-based image reconstruction is innovative.

The methodology is rigorous, with a clear pipeline from EEG signal processing to image reconstruction. The use of multiple metrics for performance evaluation ensures a comprehensive assessment of the reconstruction quality.

The paper is well-organized, with a clear presentation of the reconstruction pipeline and detailed explanations of the methods and results. The figures and tables effectively support the textual content.

**Weaknesses:**

1. The authors do not clearly state the main contribution of this paper. Since reconstructing visual images from EEG is not a novel concept and has been extensively studied in the literature, it is unclear what new contributions this paper makes to the field. Is it merely the use of more powerful generative models? If the reconstructed image quality is higher, is it because the generative models used are superior, or because more information has been decoded from the brain signals?

2. It appears that the authors only report metrics for image reconstruction and do not calculate accuracy metrics for feature decoding from EEG.

3. The authors validate the effectiveness of their method using only one EEG dataset. Given the extensive literature on similar work based on fMRI or MEG, the authors should demonstrate the superiority of their proposed method across more datasets (e.g., NSD).

4. The authors should provide a deeper analysis of why visual stimulus information (such as color, shape, texture, etc.) can be reconstructed from EEG. Does this mean that EEG already carries enough information from the primary visual cortex, or is this information merely an illusion created by the generative models?

**Questions:**

See above.

---

> ### Author Response · Authors · 2024-11-23
> **Response to qxem**
>
> Thank you for your review of our manuscript. We have responded to your written comments below, with references to added supplementary materials for further clarification.
>
> **Response to Question 1:**
>
> To the best of our knowledge, there is a gap in the literature of EEG visual reconstructions. Li et. al. 2024 is a contemporaneous work performing reconstruction with a neural network-based architecture. Though they explored the performance using different sets of time windows and electrodes, there is no direct connection between specific EEG features and model predictions. Our work not only achieves similar reconstruction performance with a much simpler method, but we also show spatiotemporal semantic-related EEG patterns linked to model predictions in a much more detailed way.
>
> Prior to THINGS-EEG2, a popular dataset for visual decoding was the EEGCVPR40 dataset (Spampinato et al. 2017), which is used by many studies from different researchers (Palazzo et al. 2018, Tirupattur et al. 2018, Deng et al. 2023, Ferrante et al. 2024, Singh et al. 2024). These may partly contribute to the notion that EEG visual reconstruction seems to be well studied. Concerns were raised about the validity of EEGCVPR40 (Li et al. 2020), particularly due to their blocked design allowing slow-changing DC drifts and noise statistics to coincide with the duration of each block. (i.e. the classifier can simply tell which class the image is from by the noise statistics because all images from that class were recorded around the same time and thus sharing the same noise statistics). Additionally, few of these studies tackled the issue of revealing the relevant features in the EEG.
>
> In the bigger picture, whether the EEG trial averages in the fully randomized order is equivalent to trial averages from consecutive presentations of the same image remains an open question. Going back in time, studies on specific visual features were scarce and a more popular topic was the joint effects between visual features and attention. This is partly due to the low Signal-to-Noise Ratio in the EEG but also the tradition in the ERP field of focusing more on attention and surprisal. Only recently, the sheer number of averages in the THINGS-EEG2 test set allows us to visualize class differentiating features even in the raw ERP for the first time.
>
> Our paper offers novel contributions that are relevant to the field. Firstly, we show several neural and latent space properties that impact various aspects of an image reconstruction. This is key for understanding the mechanisms underlying brain-to-image decoding, which are typically obscured by the black box of stable diffusion and neural network decoding algorithms. In addition, our EEG pattern visualization pipeline implements the stable unCLIP model, which provides more efficiency than the standard Versatile Diffusion CLIP encoding while still maintaining high reconstruction fidelity.
>
> **Response to Question 2:**
>
> We agree that having the retrieval performance would allow a more holistic evaluation of the model performance. Four of the reconstruction performance metrics (AlexNet-2, AlexNet-5, Inception, and CLIP) are the top-2 accuracy between the encoded embedding of the reconstructed image and the ground-truth image, as this is the standard across studies that used this set of metrics.
>
> **Response to Question 3:**
>
> We agree that it would be a great idea to apply our decoding-encoding loop to extract visual semantic maps from the NSD. Though the scope of this paper is focused on revealing the spatiotemporal patterns in the EEG that provide information about semantics and visual features, the method should be applicable to other recording modalities. fMRI should provide a much more spatially detailed and fine-grained map and potentially locate the source of the EEG patterns that we observe. Since we initially applied it to MEG at the early stage of the project, we are also planning on revisiting it and applying the spatiotemporal map technique to it. Monkey visual cortical data should be able to provide much more fine-grained RSA with which to improve ML representational models such as CLIP, and our method should be able to shed light on the corresponding population coding.

---

> > ### Comment · Reviewer_qxem · 2024-11-27
> >
> > Thank you for your reply. Although the authors demonstrate that high-quality images can be reconstructed from EEG, I still think this is mainly with the help of the diffusion model priors. The authors do not dissect which information can actually be decoded from EEG and which is not. Overall, I feel that this paper's innovation and contribution to the field is limited, and I maintain my score.

---

> > > ### Author Response · Authors · 2024-12-04
> > > **Response to qxem**
> > >
> > > This study is not meant to provide an exhaustive list of all the visual features that can or cannot be decoded from EEG. But we provide a list of latents each specializing on a general type of visual features. We provide a few manually defined features such as brightness, hue, and texturedness that are well decoded from EEG. And thanks to the linear model we find specific EEG patterns linearly contributing to those features which are described in the paper.
> > >
> > > While the quality of the reconstructions is helped by the quality of the diffusion model, this is also true for the previously published fMRI/MEG/EEG papers as well.   The mapping from the EEG to the CLIP latents is still learned and provides the guidance for the diffusion.  In our EEG swapping experiment, we clearly show that when the EEG is changed, the generated image changes in a predictable manner.
> > >
> > > Out of all the recent fMRI/MEG/EEG reconstruction work, we feel this work delves the most into the visual features.
> > >
> > > The fact that many reviewers find it surprising that linear operations on EEG can reveal visual and semantic features is part of what makes the work exciting - we were also surprised when we first saw the results and recognized the importance of the finding.   We understand the skepticism, but we provide code on github (which has been successfully used by others in our group) for full reproducibility/proof that it works. We should also emphasize that the response is based on the average response from 80 test-presentations (as with the other studies using THINGS-EEG 2) and that the traditional human electrophysiology (EEG based Cognitive Neuroscience) field is based on averaging (a linear operation) of many time-locked EEG presentations (e.g. P300, N170, Dm, N400,...).

---

> ### Author Response · Authors · 2024-11-23
> **Response to qxem (Part 2)**
>
> **Response to Question 4:**
>
> The raw EEG RSA plot reveals that the raw EEG contains relevant visual semantics information. Mapping the EEG to the CLIP representation enhances agreement with the CLIP embedding, which encodes high-level semantics well. For a more detailed exploration of the EEG features linked to low-level visual features such as brightness, color, and texture, see Section 2 in the Supplemental Material. The generative model combines the high-level semantics with the low-level visual features to synthesize a high-quality reconstruction.
>
> The argument we offer in our paper is that the visual features are all encoded in the EEG activity from visual perception, but their reconstruction is dependent upon the latent spaces used for image encoding and decoding. In other words, low-level visual features are extracted from EEG, but the diffusion model also enables high-quality reconstructions based on the high-level semantic information in the EEG. The question you pose here is an extremely relevant one, especially in terms of representation learning between brain and latent space, that warrants further investigation in our future investigation.

---

### Official Review · Reviewer_BqZF · 2024-11-03

**Soundness:** 2
**Presentation:** 2
**Contribution:** 1
**Rating:** 5
**Confidence:** 4

**Summary:**

The paper adapts an fMRI brain-to-image decoding pipeline based on linear Ridge regression and frozen image latents to EEG brain data. Linear decoders are trained per-subject and token of each latent representation (CLIP-Vision, CLIP-Text and VDVAE) on the averaged trials of each 16,540 train images (repeated 4 times each). Predicted latents on a test set of 200 images (averaged over 80 trials per image) are then fed to a Versatile Diffusion model, which generates images whose similarity to the ground truth is evaluated with different image metrics. Ablations suggest using the full EEG windows (800 ms) and all three latents yield the best performance. Additional analyses show (1) reconstructions based on PCA and ICA image features, (2) the effect of changing the layout of EEG electrodes, (3) the impact of swapping out subwindows between EEG examples on the reconstruction, and (4) spatiotemporal activation maps for three different categories of images.

**Strengths:**

* Quality: Relevant analyses and ablations are proposed (e.g. impact of averaging trials, use of the different latents, neuroscience-inspired analyses). The time-swapping experiment yields interesting results.
* Clarity: The paper is overall clear and the different analyses are explained appropriately.
* Significance: The paper shows that a simple pipeline based on linear models can achieve good performance on an image decoding task.

**Weaknesses:**

1. The novelty of the work is limited. There are no new methodological contributions as the proposed approach appears to be a direct adaptation of Ozcelik & VanRullen (2023), but on EEG rather than fMRI. The proposed analyses (reconstructions based on PCA/ICA, effect of electrode layouts, etc.) appear to be new and may be interesting from a brain decoding perspective, but their significance is limited for the wider ML/AI community.
2. The comparison to existing work is limited. The results of Li et al. (2024) and Benchetrit et al. (2024) were for cross-subject models, while results presented here are for subject-specific models only. I believe the proposed approach may not be straightforward to apply to a multi-subject setting given the use of linear models, but this would be a better point of comparison. Moreover, the direct comparison to the results on the THINGS-MEG dataset is not appropriate as the brain data itself is different (EEG vs. MEG) and the number of image repetitions is not the same (1 vs. 4 presentations of training images; 12 vs. 80 presentations of test images), see Q4.

**Questions:**

1. Why is the performance so high as compared to existing work that uses more sophisticated approaches (e.g. Li et al., 2024)? Can the authors highlight differences with other work that may explain why the proposed simpler approach works as it does?
2. What is the impact of the “shift and rescale” procedure described in lines 142-143? This would be interesting to include as an ablation too.
3. How was the regularization strength selected (line 160)? Are the models sensitive to the choice of this hyperparameter?
4. Does the proposed approach yield (or is expected to yield) similar results on the THINGS-MEG dataset? I see the shared code contains instructions on how to run on THINGS-MEG, but I don’t see matching results in the paper.
5. Figure 5: It would be interesting to see what images reconstructed from the groundtruth’s first 1000 principal components look like. This may be a better point of comparison than the groundtruth images themselves.
6. Does the data for Figures 5, 6, 7 come from subject 1 as well? Generally speaking, can the authors confirm that results (reconstructions) shown on subject 1 only generalize to other subjects too?
7. What are the + and - columns in Figure 9?

---

> ### Author Response · Authors · 2024-11-23
> **Response to BqZF**
>
> Thank you for your review of our manuscript. We have responded to your written comments below, with references to added supplementary materials for further clarification.
>
> **Response to Weakness 1:**
>
> Though some other research efforts have utilized EEG for visual percept decoding, the novelty of our work lies in the combination of our approach and investigation of the mediating role that the representational (latent) space plays in decoding neural representations of visual stimuli. Our EEG pattern analysis pipeline includes the use of the stable unCLIP model for encoding and decoding images, which achieves a similar level of performance. Further adding to its merit in the ML/AI field, unCLIP utilizes a lower-dimensional latent space than CLIP-Vision – a quality that reduces computational complexity and makes the EEG pattern analysis easier.
>
> We agree with your claim that the analyses proposed are interesting from a brain perspective, but we argue that our data demonstrates a clearer representational mapping between the brain and neural network models than other image decoding studies. As a model system for natural computation, understanding how the brain processes perceived images – and how its activity maps to neural network representations at a range of spatiotemporal scales – can inspire improvements upon existing ML/AI tools for the same tasks. We should, and will, make this benefit more clear in our rewrite of the manuscript.
>
> **Response to Weakness 2:**
>
> Li et. al. 2024 made a distinction between the in-subject model and the jointly-trained model. Though Figure 3 in that paper indicates the joint-subject training for the model, many of their performance measures (Figure 5 (c,d), 9 (a)) are reported using the in-subject training. The comparison between the in-subject and joint-subject models shows drastic differences in performance (Figure 9 and Table 8): top-5 performance drops from 55.32% to 25.36%, and top-1 performance drops from 26.13% to 8.24%. Some aren’t clearly stated but are likely to be in-subjects given the reported performance being much closer to the in-subject one: Figure 7, 8, 29, 30, 31, 32, Table 1, 3, 4, 6, 7. Given the description of “for subject 8” and their GitHub code instruction for running the reconstruction includes “subject 8”, it’s safe to assume that the reconstruction performances are in-subject only and no joint-subject reconstruction performance was provided.
>
> Benchetrit, et. al. 2024 used the model from Déffosez et. al. 2023, which learns a Participant Matrix for each participant. It is similar but more expressive than participant embeddings. The role of it is to let the model know which participant it is decoding. It is an interesting architecture that allows certain parts of the model to be the same for all subjects, but it is not predicting on unseen subjects.
>
> A previous version of our paper included a cross-subject training framework, but we didn’t include it in this submission because we wanted to focus on the EEG features of visual representations. But we have now added it to the Supplemental Material Section S.4. For our cross-subject model we used a transfer-learning approach under the assumption of a few training trials from a “target” subject and many more trials from source subjects.  We augment the small number of training trials (e.g. 3000) from the target subject with “predicted” samples obtained from linear regression from training samples from other subjects. This regressor is trained to predict the target data from other subjects’ data  (on the 3000 given samples) and is then used to generate predicted data (on the other 13540 samples).  The 3000 real samples are still the main contributor to the performance, but the 13540 provide some boost to the performance for free.

---

> ### Author Response · Authors · 2024-11-23
> **Response to BqZF (Part 2)**
>
> **Response to Question 1:**
>
> We address the high performance of a simple model in Section 1 of our added supplemental material. Put briefly, we discuss that a simple approach such as regression has theoretical backing from findings in the literature. Our argument is based on the compressibility of the CLIP latent space, evidence of neural network layers predicting increasing layers in the visual processing hierarchy in the brain, evidence of the success of linear probing of the CLIP latents for many visual classification tasks, and our own investigations that find a common structure in the representational similarity analysis plots of EEG and CLIP (compared to pixel space RSA plots).
>
> **Response to Question 2:**
>
> The shift and rescale is a simple yet effective way of avoiding poor performance caused by differing standard deviations between the predicted and ground truth embeddings. In the case of subject 1 using the unCLIP pipeline, the normalization raises the CLIP reconstruction performance from 0.780 to 0.807. See Section 1.2 in the Supplemental Material PDF for more information.
>
> **Response to Question 3:**
>
> We looked at the performance of subject 1 with regularization strengths of 0, 1, 10, 100, 1000, 10000, and 100000. Technically a more rigorous way is using cross-validation by having a couple of subjects purely for picking the regularization strengths. For Versatile Diffusion, having 0 as the regularization strength versus non-zero values makes a small  difference, but quickly tapers off from 10, 100, etc. The exact value of the regularization strengths has very low impact aside from that.
>
> **Response to Question 4:**
>
> We applied it to THINGS-MEG during the early stage of this project, and the performance was very similar to what was reported by Benchetrit et. al. 2023. But since we found that the model worked on THINGS-EEG 2, our focus shifted towards that direction because visual reconstruction working on EEG was much more unexpected given the poorer spatial resolution of EEG. The repository does contain THINGS-MEG code, but it’s not as well organized as the THINGS-EEG 2 one.
>
> **Response to Question 5:**
>
> See Figure S.20 in the Supplemental Material for the visual quality of reconstructions with the ground-truth Principal Components of the images (without using EEG). The 1000 components produce images that look visually similar to the ground-truth image but slightly blurry.
>
> **Response to Question 6:**
>
> Yes, they are. We’ll make it clear in the main document.
> Figure 3(a) shows the performance of each subject. There is nothing unique about Subject 1, who is neither the best nor the worst subject. So far we haven’t noticed any results that aren’t generalizable across subjects aside from each subject having somewhat unique EEG patterns.
>
> **Response to Question 7:**
>
> A “+” indicates that category if there is a more positive voltage there; a “-” indicates that category if there is a more negative voltage there. (i.e. they reflect where increased positive and negative activity indicates the category).  We will add that clarification to the figure caption.

---

> > ### Comment · Reviewer_BqZF · 2024-11-25
> >
> > I thank the authors for their answers.
> >
> > W1: unCLIP has been used successfully in previous brain decoding work [1], therefore the novelty brought by this methodological choice is also limited. Overall, while I see how the exploratory analysis of image representations and EEG “patterns” may be interesting to the smaller brain decoding community, it remains unclear to me how this can more generally guide the development of ML/AI tools.
> >
> > [1] Scotti, Paul S., et al. "MindEye2: Shared-Subject Models Enable fMRI-To-Image With 1 Hour of Data." arXiv preprint arXiv:2403.11207 (2024).
> >
> > W2: Thank you for the additional information. If I understand correctly, the results from Li et al. (2024) are not clearly on an across- or within-subject setup, but are likely obtained from Subject 8 in a subject-specific setting. The results from Benchetrit et al. (2024) are in a cross-subject setting (training on all subjects, testing on all subjects), and on a different brain modality (MEG). I maintain that the different evaluation settings make it difficult to compare the proposed approach to previous work. An updated version of the manuscript should match these previous settings as much as possible to provide a clear comparison point.
> >
> > Q1: Thank you for the additional discussion. However I believe this does not answer why a linear regression may work better than a model with bespoke inductive biases (e.g. the Transformer from Li et al. 2024, or the ConvNet from Benchetrit et al. 2024), whose hypothesis space may also contain a (similar) simple linear mapping.
> >
> > Q2-7: Thank you for taking the time to address these additional questions.
> >
> > Overall, I increase my score to a 5, but still believe as Reviewer P37j that the limited novelty and technical contribution of the  manuscript calls for a more thorough rework or refocus of the paper.

---

### Official Review · Reviewer_ef9S · 2024-11-04

**Soundness:** 3
**Presentation:** 2
**Contribution:** 2
**Rating:** 5
**Confidence:** 4

**Summary:**

The paper presents Perceptogram, a novel approach for reconstructing viewed images from EEG recordings. The method utilizes a linear decoder to map EEG signals to image latents, leveraging latent diffusion guided by CLIP embeddings for image reconstruction. The study also explores reconstructions using PCA and ICA components capturing luminance and hue-related information. The linear model provides interpretable EEG features for differentiating semantic categories, and the results demonstrate state-of-the-art quantitative reconstruction performance.

**Strengths:**

The paper uses a liner regressor to correlate the EEG space and VAE/CLIP space and achieves good performance in EEG-based image recognition. Meaningful analysis has been conducted to provide insights, including ICA and PCA features, scalp-asymmetry effect, and spatial-temporal semantic maps.

**Weaknesses:**

The linear model approach is a conventional method for brain visual reconstruction, offering high interpretability. However, further clarification is needed to enhance the reliability of this work:
1. How effectively does the regressor perform in mapping EEG embeddings to CLIP embeddings? Could specific visual components represented by different tokens show higher correlations with EEG features?
2. The study lacks key conclusions derived from performance metrics and analysis on the nature of visual information learned from EEG signals.

**Questions:**

1. Why not utilize whole-brain data to analyze spatial patterns? High-level information is often associated with the temporal cortex, where the EEG channels have been excluded.
2. Could you compare the contributions of the VAE, CLIP-vision, and CLIP-text components? Which component plays a more significant role in image reconstruction, and what types of features are critical? It may be useful to show reconstruction results for each module individually.
3. What insights can be drawn from Figure 3(d)? It seems that different cases yield very similar performance.
4. How was the model trained, and what objective functions were used? Did training mainly involve constraining the three regressors?
5. How were the error bars in Figure 3(a) generated if all subjects were included?
6. What information can be inferred from the spatiotemporal semantic map? Currently, the map only shows varied patterns across categories. More substantial evidence is needed to establish consistency with established neuroscience knowledge.

---

> ### Author Response · Authors · 2024-11-23
> **Response to ef9S**
>
> Thank you for your review of our manuscript. We have responded to your written comments below, with references to added supplementary materials for further clarification.
>
> **Response to Weakness 1:**
>
> The topic of the _effectiveness of linear models_ as well as _specific visual features_ is of great interest to us and various related questions were raised by many reviewers. Therefore we have created a Supplemental Material PDF to be able to fully respond with visualizations. Specifically, section S1 addresses the effectiveness of linear modeling for mapping EEG to the CLIP latent space, and section S2 addresses the specific visual components in the EEG.
>
> **Response to Weakness 2:**
>
> We briefly address in our discussion section the performance gain compared to other brain-to-image reconstruction studies, as well as the implications of our visual feature findings. This, however, is further revisited in our added Supplemental Materials (S2), in which we detail the effects of different latent spaces on 1) the visual features of the reconstructed images and 2) the encoding patterns of EEG data. In that section, we conclude that the latent space used for encoding impacts different visual features in the processing hierarchy, from low (e.g., PCA) to high (e.g., CLIP).
>
> **Response to Question 1:**
>
> We initially tried using all 64 channels from the whole brain but found the visual area performed slightly better than using the whole brain. So the subsequent analyses were all performed on the 17 posterior channel preprocessed data.
>
> **Response to Questions 2 and 3:**
>
> Figure 3(d) compares the relative contributions of VDVAE, CLIP-Vision and CLIP-Text to the reconstruction performance. To provide a sense of scale for those metrics, Figure 2(b) has performance from “0 ms” which indicates the lowest possible performance by using EEG signal prior to trial onset.
>
> 	“It may be useful to show reconstruction results for each module individually.”
>
> We have just added the VDVAE reconstructions to the Appendix of the main paper and CLIP-Vision-only and CLIP-Text-only reconstructions to the Supplemental Material (due to the file size limit for the main paper).
>
> **Response to Question 4:**
>
> Yes, the training only involves constraining the required number of regressors. We used Ridge Regression with regularization strength of 1000 for VDVAE and CLIP-Vision (all 257 of them) and 10000 for CLIP-Text (all 77 of them). For Versatile Diffusion, each token has its own regressor, so there are in total 257 (CLIP-Vision) + 77 (CLIP-Text) + 1 (VDVAE) = 335 regressors to constrain. For unCLIP, because there is only 1 token for its variant of CLIP, there is only one regressor with regularization strength of 1000, plus another regressor for VDVAE.
>
> **Response to Question 5:**
>
> The error bars in Figure 3(a) represent the standard deviation of 7 different generations from 7 different random seeds for Versatile Diffusion.
>
> **Response to Question 6:**
>
> Linking spatiotemporal semantic maps to neuroscience knowledge is non-trivial due to how removed the electrodes are from the brain tissues. The next best proxy to which EEG can be related is fMRI. Jain et al. 2023 explored selectivity for food in human fMRI data. Figure 2 in that paper showed a more lateral activation for food, potentially linked to the lateral preference for food in the EEG patterns that we found. Connolly et al. 2012 found animate-inanimate selectivity in the Lateral Occipital Complex, which is more posterior than the region selective for food. The caveat here is that the voltages recorded on each electrode do not necessarily indicate activity directly under that electrode; it also depends on the direction of the dipole and the way volume conduction affects the electric field. An fNIRS dataset of a similar experiment design may help elucidate the relationship between EEG voltages and localities of the cortical activation sources.
>
> Another way to test the robustness of the spatiotemporal semantic map is by running variations of the EEG experiment (e.g. changing the style of the visual stimulus, showing the stimuli as videos or 3D objects, etc) and see if the spatiotemporal map still holds.

---

> > ### Comment · Reviewer_ef9S · 2024-11-27
> > **Response to authors**
> >
> > Thanks for replying to my comments. Some of my concerns have been addressed. Overall, this paper's analysis is interesting but more experiments to show the information we obtained from the brain recordings would benefit the contributions. I'll keep my scores.

---

### Author Response · Authors · 2024-12-04
**Global response**

We thank you for your helpful input. We would like to give the following closing argument.

The fact that many reviewers find it surprising that linear operations on EEG can reveal visual and semantic features is part of what makes the work exciting.  At the same time, it is not magic; the reconstruction is based on the average response from 80 test-presentations (as with the other studies using THINGS-EEG 2) and the traditional human electrophysiology (EEG based Cognitive Neuroscience) field is also based on averaging (a linear operation) of many time-locked EEG presentations (e.g. P300, N170, Dm, N400,...).   The surprising aspect is the extraction of fairly detailed semantic and visual information.  We should emphasize that while the realism of the generated images benefits from the Diffusion model (as is true for the fMRI and non-linear EEG models also), the visual and semantic match between the shown image and the reconstructed image can only come from information extracted from the EEG.  We have demonstrated that this information can be extracted with regularized linear regression mapping to appropriate latent spaces.

We feel that our work may be being held to a higher standard  than methods that employ complex neural network architectures.  We believe in Occam’s razor, and that more complex methods should justify their added complexity - we show that for the same and slightly better performance, the added complexity is not needed - this finding by itself is a contribution and can help simplify and reduce computation/environmental impacts in future work.  Our approach, using simple ridge regression, offers both interpretability and performance without relying on unnecessary complexity.   Specifically we have developed the decoding-encoding loop for visualizing the features in the EEG that the linear regression is extracting in the map to the latent space.  Different latent spaces reveal different coding features in the EEG and this explicit linkage enhances interpretability and provides a precise framework for identifying meaningful EEG patterns that our model has extracted.

---

### Meta-Review · Area_Chair_sN1F · 2024-12-19

**Metareview:**

This work presents a method for the reconstruction of observed images from EEG recordings using a linear decoder that maps EEG data to image latents while adopting latent diffusion guided by CLIP embeddings for the ctual reconstruction. The linear model provides interpretable EEG features for differentiating semantic categories, and creates spatio-temporal semantic maps reflecting the temporal evolution of class-relevant semantic information.

The paper has several interesting characteristics, which were appreciated by the reviewers such as clarity of the presentation, significance of the proposed approach, good experimental analysis.

Reviewers also raised several remarks such as the weak novelty, insufficient proof of the method's reliability (which impacts the experimental analysis) and lack of sufficient comparisons with existing works, the lack of clear conclusions from the measured performance metrics, and other specific questions regarding the method and the experimental results.

Originally receiving negative scores (3, 5, 5, 5), the situation does not improve much after rebuttal (5, 5, 5, 5), still globally resulting in below-threshold ratings. Authors provided extensive rebuttal to the many raised remarks, which have satisfied reviewers only partially, while leaving still open some doubts that prevented a net appreciation of the work.

In these conditions, this paper cannot be considered acceptable for publication to ICLR 2025.

**Additional Comments On Reviewer Discussion:**

See above

---

### Decision · Program_Chairs · 2025-01-22

Reject